# Endonuclease G promotes autophagy by suppressing mTOR signaling and activating the DNA damage response

Wenjun Wang[1,2,3,10], Jianshuang Li[2,3,10], Junyang Tan[3], Miaomiao Wang[3], Jing Yang[3], Zhi-Min Zhang [4], Chuanzhou Li[5], Alexei G. Basnakian[6], Hong-Wen Tang[7,8], Norbert Perrimon[8,9] & Qinghua Zhou [1,2,3 ✉]

Endonuclease G (ENDOG), a mitochondrial nuclease, is known to participate in many cellular processes, including apoptosis and paternal mitochondrial elimination, while its role in autophagy remains unclear. Here, we report that ENDOG released from mitochondria promotes autophagy during starvation, which we find to be evolutionarily conserved across species by performing experiments in human cell lines, mice, *Drosophila* and *C. elegans*. Under starvation, Glycogen synthase kinase 3 beta-mediated phosphorylation of ENDOG at Thr-128 and Ser-288 enhances its interaction with 14-3-3γ, which leads to the release of Tuberin (TSC2) and Phosphatidylinositol 3-kinase catalytic subunit type 3 (Vps34) from 14-3-3γ, followed by mTOR pathway suppression and autophagy initiation. Alternatively, ENDOG activates DNA damage response and triggers autophagy through its endonuclease activity. Our results demonstrate that ENDOG is a crucial regulator of autophagy, manifested by phosphorylation-mediated interaction with 14-3-3γ, and its endonuclease activity-mediated DNA damage response.

[1] The First Affiliated Hospital, Jinan University, Guangzhou, Guangdong 510632, China. [2] Zhuhai Institute of Translational Medicine Zhuhai People's Hospital Affiliated with Jinan University, Jinan University, Zhuhai, Guangdong 519000, China. [3] The Biomedical Translational Research Institute, Faculty of Medical Science, Jinan University, Guangzhou, Guangdong 510632, China. [4] International Cooperative Laboratory of Traditional Chinese Medicine Modernization and Innovative Drug Development of Chinese Ministry of Education (MOE), College of Pharmacy, Jinan University, Guangzhou, Guangdong 510632, China. [5] Department of Medical Genetics, School of Basic Medicine and Tongji Medical College, Huazhong University of Science and Technology, Wuhan 430030, China. [6] Department of Pharmacology & Toxicology, University of Arkansas for Medical Sciences, Little Rock, AR 72205, USA. [7] Program in Cancer and Stem Cell Biology, Duke-NUS Medical School, 8 College Road, Singapore 169857, Singapore. [8] Department of Genetics, Blavatnik Institute, Harvard Medical School, Boston, MA 02115, USA. [9] Howard Hughes Medical Institute, Boston, MA 02115, USA. [10]These authors contributed equally: Wenjun Wang, Jianshuang Li. ✉email: gene@email.jnu.edu.cn

Autophagy is a lysosome-dependent degradation process that protects cells under conditions of starvation, stress or infection[1]. The process of autophagy is executed by ATG proteins (ULK1, Belcin1, ATG3/5/7/12, LC3, etc.) and is regulated by several signaling pathways, including mTORC1 and AMPK, as well as by epigenetic changes[2–4]. mTOR is one of the most important pathways that negatively regulates autophagy in mammalian cells[1]. mTORC1 inhibits ULK1-dependent autophagy by phosphorylating ULK1 at serine 757[5]. mTORC1 also suppresses autophagy by phosphorylating and inhibiting nuclear translocation of the transcription factor EB (TFEB), which promotes the transcription of lysosomal biogenesis and the autophagic process[6].

It has been shown that 14-3-3 proteins play complicated roles in autophagy. Under normal growth conditions, 14-3-3 proteins bind and regulate TSC2 (Tuberin) and PRAS40 (Proline-rich AKT1 substrate 1), which can activate mTORC1, resulting in the inhibition of autophagy[7,8]. During nutrient starvation, 14-3-3 proteins bind to Raptor and ULK1 to promote autophagy initiation[9,10]. In addition, Vps34 (phosphatidylinositol 3-kinase catalytic subunit type 3), which plays an important role in autophagosome formation, directly interacts with 14-3-3 proteins under normal growth conditions in a phosphorylation-dependent manner[11]. Furthermore, Vps34 lipid kinase activity is increased following dissociation from 14-3-3 during starvation-induced autophagy or 14-3-3ζ downregulation[11].

Endonuclease G (ENDOG) is encoded by the nuclear gene *ENDOG*, and mainly located in the mitochondrial intermembrane space[12]. ENDOG can be released from the mitochondria and translocated to the nucleus where it induces fragmentation of GC-rich genomic DNA, causing apoptosis[12]. ENDOG also plays important roles in mtDNA biogenesis[13], as well as in human diseases such as cardiac hypertrophy, Parkinson's disease, and obesity[14–16]. Our previous study showed that during fertilization in *C. elegans*, paternal ENDOG relocates from the mitochondrial intermembrane space to the matrix, where it digests the paternal mtDNA and promotes paternal mitochondrial elimination through the maternal autophagy machinery[17]. However, the role of ENDOG in regulating general autophagy remains unclear.

In this work, we investigate the function and underlying mechanisms of ENDOG in regulating general autophagy. By identifying ENDOG as a phosphorylation substrate of GSK-3β (glycogen synthase kinase 3 beta) and the interaction between ENDOG and 14-3-3γ, we reveal that GSK-3β mediated phosphorylation of ENDOG enhances its interaction with 14-3-3γ, leading to the release of TSC2 and Vps34 from 14-3-3γ, and eventually promotes mTOR pathway suppression and autophagy initiation. Moreover, ENDOG also promotes autophagy through its endonuclease activity-mediated DNA damage response. Our study shed light on the molecular mechanisms for the physiologic function of ENDOG during autophagy.

## Results

### ENDOG promotes autophagic flux in hepatocytes

Our previous results showed that CPS-6, a homolog of human mitochondrial endonuclease G (ENDOG), functions as a paternal mitochondria degradation factor through autophagy machinery upon fertilization in *C. elegans*[17]. We wondered whether ENDOG regulates general autophagy. Thus, we performed gain- and loss-of-function experiments to examine the role of ENDOG in autophagy. In two human liver cell lines (L02 and HepG2), ENDOG increased LC3B-II accumulation, decreased the expression of the autophagy substrate SQSTM1, and promoted autophagosome formation (visualized as GFP⁻LC3 puncta) (Fig. 1a–d and Supplementary Fig. 1a, b). However, the transcription of

autophagy-related genes was not affected by ENDOG over-expression or knockout (Supplementary Fig. 1c, d). Furthermore, loss of ENDOG significantly repressed the expression and phosphorylation of many core ATG proteins (Supplementary Fig. 1e, f). These data suggested that ENDOG promoted autophagy in hepatocytes.

Next, we investigated whether ENDOG regulates the autophagic flux in hepatocytes. By using the mCherry-GFP-LC3 reporter system, we found that the number of autolysosomes (mCherry⁺/GFP⁻, red puncta) were increased by ENDOG overexpression upon both normal and autophagy induction (starvation or rapamycin treatment) condition. Consistently, accumulation of autophagosomes (mCherry⁺/GFP⁺, yellow puncta) were also increased in ENDOG-overexpressing cells under chloroquine (CQ, an inhibitor of autophagosome-lysosome fusion) treatment (Supplementary Figs. 2 and 3a, b). Additionally, ENDOG promoted LC3B-II accumulation and SQSTM1 degradation under the starvation treatment (Supplementary Fig. 3c). To further confirm the role of ENDOG in the autophagic flux, we used the CRISPR/Cas9 system to construct the ENDOG knockout cell line. We found that ENDOG knockout significantly repressed the autolysosomes (red puncta) formation under both normal and stressed conditions (starvation, rapamycin, and CQ treatments). Similarly, loss of ENDOG reduced the autophagosome (yellow puncta) numbers following the CQ treatment compared with the wild-type cells (Fig. 1e–g). Moreover, the electron microscopic results showed that loss of ENDOG significantly reduces the number of autophagic vesicles (autolysosomes + autophagosomes) under the BafA1 (an inhibitor of vacuolar type H+ -ATPase (V-ATPase)) treatment (Fig. 1h, i). Knocking out ENDOG repressed starvation-induced LC3-II accumulation and SQSTM1 degradation (Fig. 1j, k). Furthermore, loss of ENDOG dramatically repressed the LC3B-II accumulation under the BafA1 treatment which prevents the acidification of lysosomes (Fig. 1l, m). Together, these results demonstrate that ENDOG promotes autophagic flux.

### Loss of ENDOG suppresses starvation-induced autophagy in multiple species

We further investigated whether ENDOG promotes autophagy in vivo and whether this induction ability is conserved in different species. In mice, compared to *Endog*⁺/⁻ (Ctrl), liver from *Endog*⁻/⁻ (KO) mice showed decreased LC3B accumulation, increased SQSTM1 expression, and reduced the number of autophagic vesicles (indicated by the electron microscopic images) under starvation treatment (Fig. 2a–d). In *Drosophila*, we detected reduced mCherry-Atg8a (LC3 homolog) puncta in ENDOG-knockdown fat body cells (equivalent to human liver), compared to those observed in adjacent wild-type cells under starvation condition (Fig. 2e). In *C. elegans*, loss of the *ENDOG* homolog *cps-6* significantly decreased the level of cleaved GFP, which serves as a degradation product in autolysosomes[18], under both normal and starvation conditions (Fig. 2f). Moreover, loss of *cps-6* suppressed the number of GFP::LGG-1 (LC3 homolog) dots, which was observed in the seam cells and pharyngeal muscle upon starvation (Fig. 2g–h and Supplementary Fig. 4a). These results suggest that ENDOG-promoted autophagy is conserved across species, including mouse, *Drosophila*, and *C. elegans*.

### ENDOG promotes autophagy partially by suppressing mTOR signaling

To explore the mechanism responsible for the induction of autophagy by ENDOG, we examined involvement of the mTOR signaling pathway, which is a well-known negative regulator of autophagy[1]. In L02 cells, loss of ENDOG increased phosphorylation levels of both mTOR (Ser2448) and its

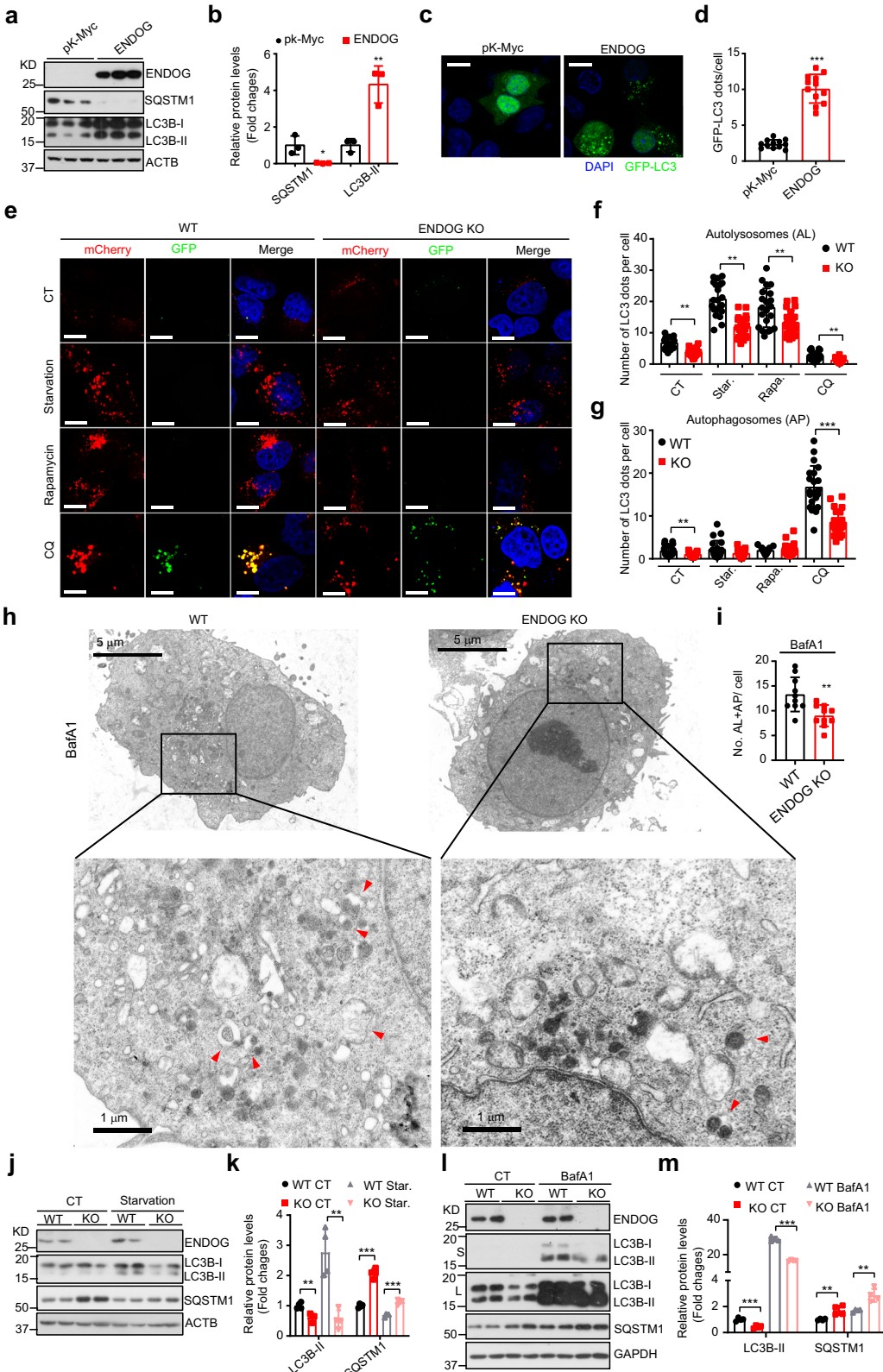

substrates (p70S6K-Thr389, ULK1-Ser757, and 4EBP1-Thr37/46) (Fig. 3a, b). Consistently, in other hepatoma cell lines, knockdown of ENDOG increased phosphorylation of mTOR and ULK1, as well as suppressed autophagy under starvation (Supplementary Fig. 4b–e). In vivo, compared to that in the *Endog*[+/−] mouse livers, the phosphorylation of mTOR-Ser2448, ULK1-Ser757, and 4EBP1-Thr37/46 were significantly increased in *Endog*[−/−] mouse

livers (Fig. 3c, d). These data indicate that ENDOG may promote autophagy through suppressing the mTOR pathway.

To further determine whether ENDOG-mediated autophagy promotion is mTOR-dependent, we used a constitutively active form of RHEB (RHEB[Q64L]) to continuously activate mTOR[19]. As a result, compared to the empty vector controls, RHEB[Q64L] partially restored mTOR activity, as shown by the increased

**Fig. 1 ENDOG promotes autophagic flux in hepatocytes. a, b** Western blots (**a**) and quantitative results (**b**) of autophagy-related proteins in HepG2 cells transiently transfected with ENDOG or pk-Myc (empty vector) for 48 h ($n = 3$ biologically independent samples, data are presented as mean values ± SD, *$p < 0.05$; **$p < 0.01$). **c, d** Representative images (**c**) and quantitative results (**d**) of GFP-LC3 puncta in ENDOG- or pk-Myc-transfected HepG2 cells (48 h after transfection; scale bar = 10 μm; $n = 75$–100 independent cells examined over three independent experiments; data are presented as mean values ± SD, ***$p < 0.001$). **e–g**. Detection and quantification of autophagic flux with the mCherry-GFP-LC3 reporter in wild-type and ENDOG-KO cell lines under indicated treatments (Starvation for 6 h; rapamycin: 1 μM for 6 h, CQ: 50 μM for 6 h). Yellow puncta, autophagosomes (mCherry$^+$/GFP$^+$); red puncta, autolysosomes (mCherry$^+$/GFP$^-$); (scale bar = 10 μm; $n = 100$ independent cells examined over three independent experiments; data are presented as mean values ± SD; **$p < 0.01$, ***$p < 0.001$). **h, i** Representative electron microscopic images (**h**) and quantitative results (**i**) of autophagic vesicles in wild-type or ENDOG knockout cells after treated with 100 nM BafA1 for 6 h (red arrow: autophagic vesicle, AL: autolysosomes, AP: autophagosomes; $n = 10$ independent cells; data are presented as mean values ± SD; **$p < 0.01$). **j–m** Western blots and quantitative results of LC3B, SQSTM1 and ENDOG in wild-type cells (WT), ENDOG knockout (KO) cells upon starvation (**j, k**) and BafA1 (**l, m**) treatments ($n = 4$ biologically independent samples) (scale bar = 10 μm; data are presented as mean values ± SD, **$p < 0.01$; ***$p < 0.001$). Source data are provided as a Source data file.

phosphorylation of ULK1 and 4EBP1(Fig. 3e, f). Additionally, RHEB$^{Q64L}$ decreased LC3B-II accumulation and autophagosome formation (GFP-LC3 puncta) in ENDOG-overexpressing cells (Fig. 3e–h). It should be noted that LC3B-II accumulation and the number of GFP-LC3 puncta in ENDOG-overexpressing cells were still more than that of wild-type cells with RHEB$^{Q64L}$ overexpression (Fig. 3e–h). Taken together, these data suggest that ENDOG promotes autophagy partially through suppressing mTOR signaling.

**ENDOG suppresses the mTOR pathway by interacting with 14-3-3γ.** To determine the mechanism underlying ENDOG-mediated suppression of the mTOR pathway, we performed an immunoprecipitation (IP)-mass spectrometry analysis of proteins associated with ENDOG. One 14-3-3 family member, 14-3-3γ, which was shown to regulate mTOR activity, was identified in the mass spectrometry results (Supplementary Fig. 5a). Co-IP experiments in ENDOG- or 14-3-3γ- overexpressing L02 cells confirmed the interaction of 14-3-3γ with ENDOG (Supplementary Fig. 5b). The immunofluorescence staining suggested that 14-3-3γ also colocalizes with ENDOG (Supplementary Fig. 5c).

Given that 14-3-3 proteins may play a role in autophagy via protein–protein interactions[9], we wondered whether ENDOG affects the interaction of 14-3-3γ with other proteins that regulate autophagy. Indeed, we found that overexpression of ENDOG repressed the interaction between TSC2/Vps34 and 14-3-3γ (Fig. 3i). Moreover, endogenous co-IP experiments showed that starvation treatment slightly enhanced the endogenous interaction between ENDOG and 14-3-3γ, while weakened the interaction of TSC2/Vps34 with 14-3-3γ (Fig. 3j). We then examined if 14-3-3γ can reverse ENDOG-induced autophagy. Overexpressing 14-3-3γ re-activated the mTOR pathway, reduced LC3B-II accumulation and SQSTM1 degradation, and decreased autophagosome formation in ENDOG-overexpressing cells (Fig. 3k–m and Supplementary Fig. 6). These data demonstrate that ENDOG competitively binds to 14-3-3γ, which releases TSC2 (a negative regulator of mTOR) and Vps34 from 14-3-3γ, resulting in mTOR pathway suppression and autophagy initiation.

**GSK-3β-mediated phosphorylation of ENDOG enhances its interaction with 14-3-3γ.** 14-3-3 proteins are known to act mainly as phosphorylation readers by binding to the general consensus sequence RXX**pS/T**XP of target proteins, which leads to their alternation in modification, intercellular localization, and activity[20]. We wondered whether 14-3-3γ binds to ENDOG via the p-S/T motif. Using the 14-3-3-Pred webserver, we predicted 14-3-3 binding motifs in ENDOG with the highest score at Serine-288 of ENDOG (Supplementary Table 1). The interactive

docking prediction also suggests that phosphorylated ENDOG at Threonine-128 and Serine-288 should be capable of forming hydrogen bonds with the Serine-59 and Glutamic Acid-18 of 14-3-3γ, respectively (Supplementary Fig. 7a, b).

Thus, we mutated Threonine-128 and Serine-288 of ENDOG to Alanine or Asparate (T128/S288 to A128/A288; T128/S288 to D128/D288), respectively, to mimic the inactive and the active form. As a result, the A128/A288 (AA) form of ENDOG suppressed the interaction between ENDOG and 14-3-3γ, which is enhanced by the D128/D288 (DD) form (Fig. 4a, b). However, individual mutations (T128D or S288D) alone had no effect on their interactions (Supplementary Fig. 7c). Compared to the wild-type ENDOG and ENDOG-DD, overexpression of ENDOG-AA has less autophagosome numbers and LC3B-II accumulation under both the control and BafA1 treatment (Fig. 4c, d and Supplementary Fig. 8). Moreover, ENDOG-AA lost the ability to suppress mTOR pathway to promote LC3B-II accumulation and SQSTM1 degradation (Fig. 4e, f). Collectively, these results suggest that dual phosphorylation of ENDOG at T128 and S288 is necessary for its interaction with 14-3-3γ and the induction of autophagy.

Next, we asked which kinase phosphorylates ENDOG at T128 and S288. Online prediction (Group-based Prediction System Web Server and NetPhos 3.1 Server) indicated that AKT and GSK-3β are the possible candidates. Surprisingly, AKT over-expression significantly decreased ENDOG phosphorylation (Supplementary Fig. 9). AKT is able to phosphorylate GSK-3β and block its activity, suggesting that GSK-3β may be more likely to be the kinase for ENDOG. Indeed, we found that overexpression of the wild-type GSK-3β, but not the kinase dead-mutant (GSK-3β-K85A) significantly increased ENDOG phosphorylation (Fig. 4g, h). Moreover, the interaction between ENDOG and 14-3-3γ was enhanced in GSK-3β-overexpressing cells (Fig. 4g, h). GSK-3β-mediated ENDOG phosphorylation and subsequent interaction between ENDOG and 14-3-3γ were abolished when the inactive form (AA) of ENDOG was introduced (Fig. 4i, j). Furthermore, we confirmed that GSK-3β phosphorylates ENDOG by in vitro phosphorylation experiment (Fig. 4k). Taken together, these results indicate that GSK-3β is a likely kinase for the phosphorylation of ENDOG, which enhances the interaction between ENDOG with 14-3-3γ.

**ENDOG promotes autophagy by activating the DNA damage response.** Given the fact that ENDOG-induced autophagy has only been partially restored in the presence of RHEB$^{Q64L}$, which continuously activates mTOR (Fig. 3e–h), we reasoned that an additional pathway might exist in ENDOG-promoted autophagy. ENDOG is a nuclease that cleaves DNA during apoptosis[12]. DNA damage response is reported as an early event during starvation[21,22] and plays an important role in autophagy[23].

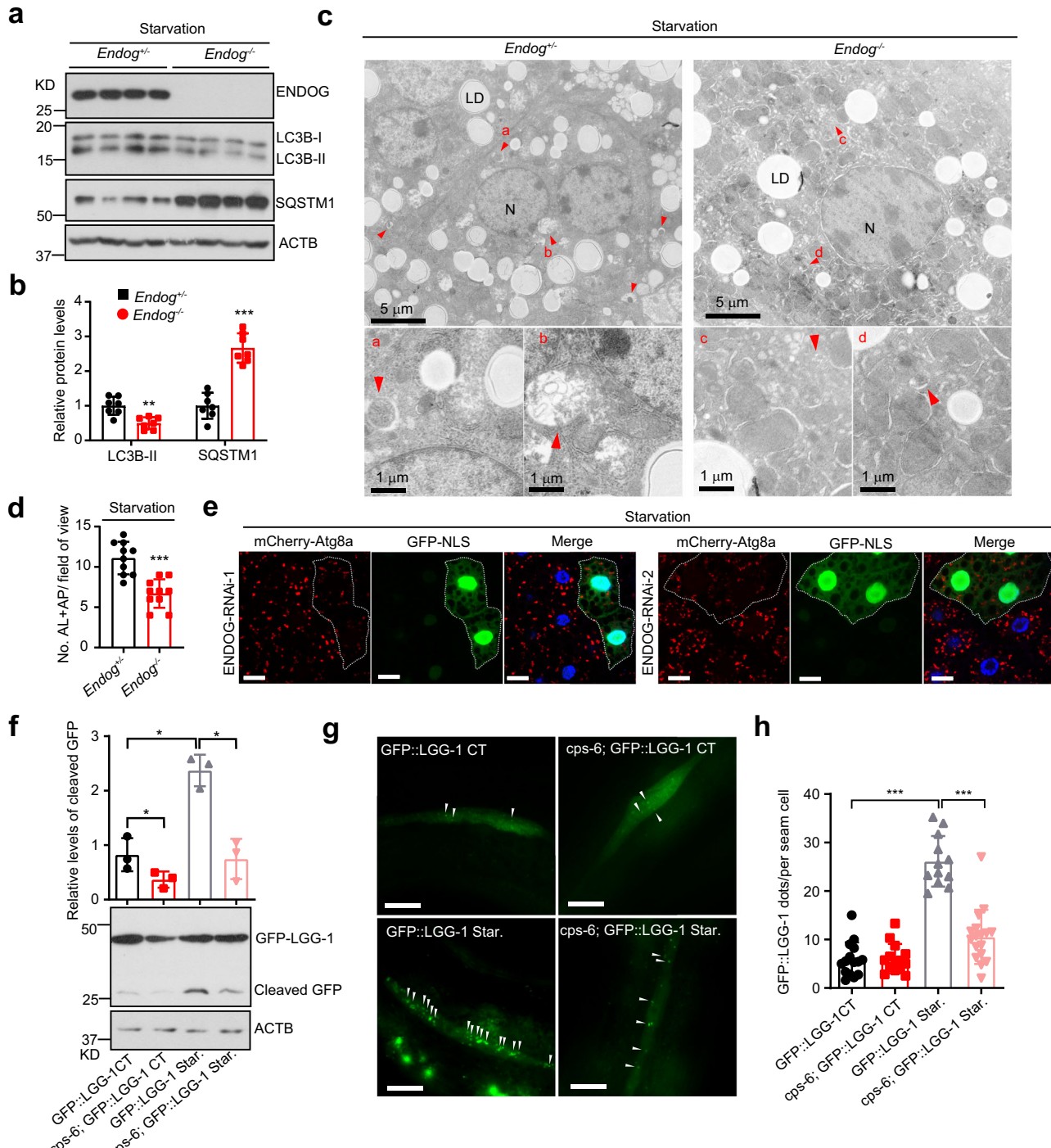

**Fig. 2 Loss of ENDOG represses starvation-induced autophagy in different species. a, b**. Western blots and quantification of LC3B and SQSTM1 in *Endog*$^{+/-}$ or *Endog*$^{-/-}$ mouse livers after starvation for 24 h (*n* = 7 biologically independent animals; data are presented as mean values ± SD; **$p$ < 0.01, ***$p$ < 0.001). **c, d** Representative electron microscopic images and quantification of autophagic vesicles in *Endog*$^{+/-}$ or *Endog*$^{-/-}$ mouse livers after starvation for 24 h (LD: lipid drop; N: nuclear; red arrow: autophagic vesicle; *n* = 10 independent fields, data are presented as mean values ± SD, ***$p$ < 0.001). **e** Representative images show that ENDOG knockdown decreased autophagosome accumulation (mCherry-Atg8a puncta) in *Drosophila* fat body cells after starvation for 4 h (GFP-NLS-labeled cells circled by dotted line express RNAi targeting ENDOG. Cells outside the circled dotted line are wild-type and used as controls; scale bar = 10 μm). **f** Representative western blots of GFP-LGG-1 and ACTB in control (GFP::LGG-1) and ENDOG loss function mutant (*cps-6*; GFP::LGG-1) *C. elegans* after starvation for 4 h. Graph, quantification of cleaved-GFP (correspond to a product of degradation in autolysosomes) (*n* = 3 biologically independent experiments, data are presented as mean values ± SD, *$p$ < 0.05). **g, h** Representative images (**g**) and quantification (**h**) of GFP-LGG-1 puncta per seam cell in *C. elegans* (white arrows: GFP-LGG-1 puncta; *n* = 10–15 biologically independent *C. elegans*, scale bar = 10 μm; data are presented as mean values ± SD, ***$p$ < 0.001). Source data are provided as a Source data file.

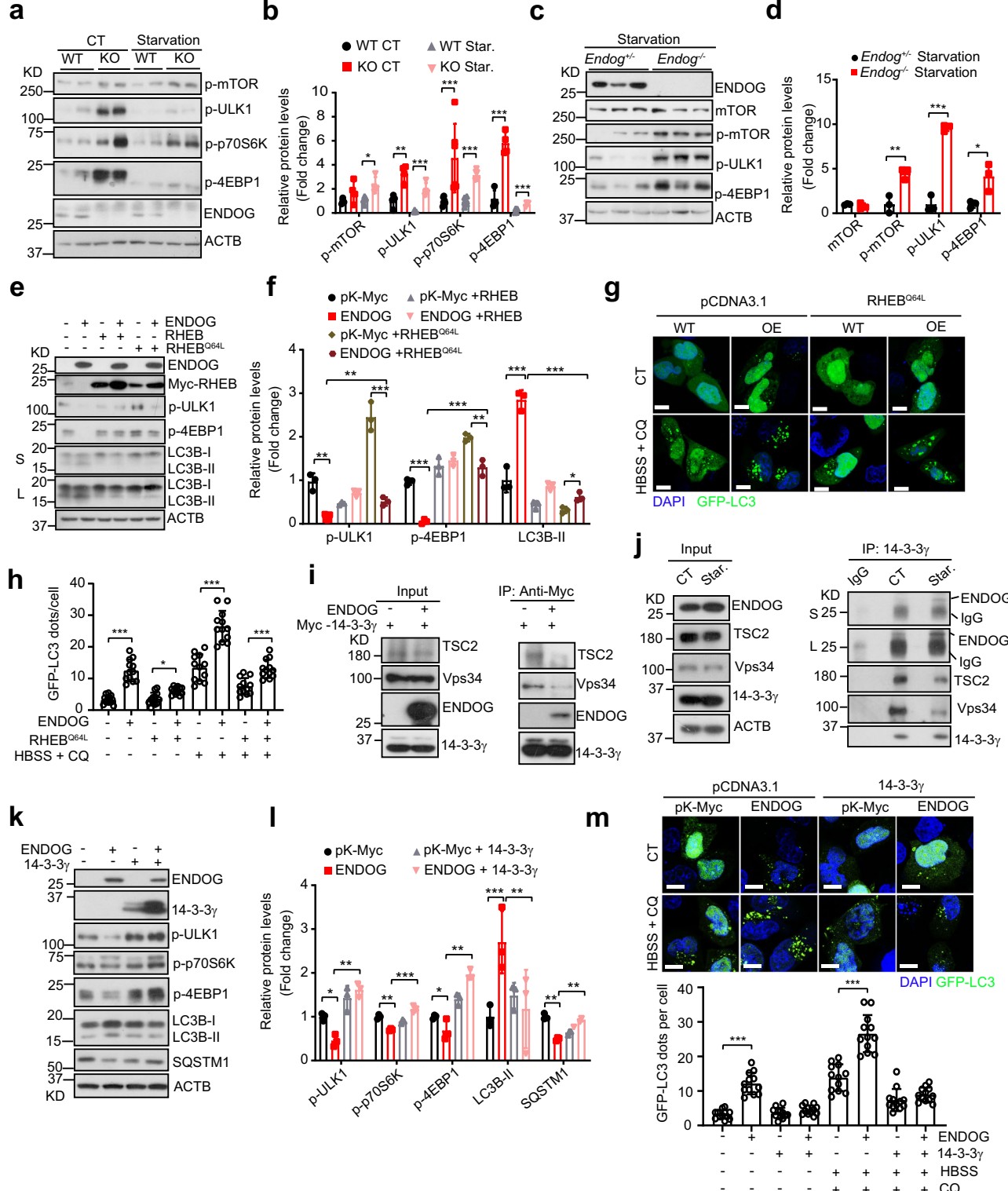

Thus, we wondered whether ENDOG affects DNA damage during starvation. The comet assay showed the existence of DNA damage both in the wild-type and ENDOG knockout cells at the early time points of starvation treatment. Compared to the wild-type cells, loss of ENDOG significantly repressed starvation-induced DNA damage (Fig. 5a–c). The p-H2A.X foci staining also showed that ENDOG knockout cell have less DNA damage during the starvation (Fig. 5d, e). These data suggested that loss of ENDOG repressed starvation-induced DNA

damage. DNA damage, an early event of starvation-induced autophagy, is mediated by PARP-1/AMPK or ATM/CHK2 pathway[21,22]. Here, we found that ENDOG knockout repressed the expression and activation of PARP-1 under normal conditions (Fig. 5f, g). The starvation-induced activation of AMPK and the repression of mTOR were blocked by the loss of ENDOG (Fig. 5f, g). Besides, the activation of CHK1 and CHK2 was also repressed in the ENDOG knockout cells, which eventually caused less autophagy (Fig. 5h, i). These data

**Fig. 3 ENDOG promotes autophagy by suppressing the mTOR pathway through its interaction with 14-3-3γ. a**, **b** Representative western blots and quantitative results of p-mTOR and mTOR's substrates (p-ULK1, p-p70S6K, and p-4EBP1) in wild-type and ENDOG knockout L02 cells following the starvation for 6 h (n = 4 independent samples, data are presented as mean values ± SD, *p < 0.05; **p < 0.01; ***p < 0.001). **c**, **d** Representative western blots and quantitative results of the indicated mTOR signaling pathway proteins in *Endog*[+/−] and *Endog*[−/−] mouse liver after starvation for 24 h (n = 3 independent animals, data are presented as mean values ± SD, *p < 0.05; **p < 0.01; ***p < 0.001). **e**, **f** Representative western blots and quantitative results of the indicated mTOR signaling pathway proteins in ENDOG and RHEB/RHEB[Q64L] co-expressed L02 cells (S: short time exposure; L: long time exposure; n = 3 independent experiments, data are presented as mean values ± SD, *p < 0.05; **p < 0.01; ***p < 0.001). **g**, **h** Representative immunofluorescence images of GFP-LC3 puncta (**g**) and respective quantitative results (**h**) in ENDOG and RHEB[Q64L] co-overexpressed L02 cell under normal and HBSS + CQ conditions (WT: wild-type; OE: ENDOG overexpressed; RHEB[Q64L] transiently transfected for 48 h; starvation for 6 h; CQ: 50 μM for 6 h; scale bar = 10 μm; n = 100 independent cells examined over 3 independent experiments; data are presented as mean values ± SD, *p < 0.05; ***p < 0.001). **i** Co-IP experiments showed that ENDOG overexpression decreases interactions between 14-3-3γ with TSC2/Vps34 (L02 ENDOG-KO cell co-overexpressed Myc-14-3-3γ with Flag-ENDOG or empty vector for 48 h; biological repeated three times). **j** Endogenous Co-IP experiments showed that starvation enhances the interaction between ENDOG and 14-3-3γ but weaken the interaction of TSC2/Vps34 with 14-3-3γ (in L02 WT cell; Star.: starvation for 12 h; S: short time exposure; L: long time exposure; biological repeated three times). **k**, **l** Representative western blots and quantitative results of the mTOR pathway and autophagy-related proteins in ENDOG or 14-3-3γ overexpressing L02 cells (ENDOG or 14-3-3γ transiently transfected for 48 h; n = 3 independent experiments; data are presented as mean values ± SD, *p < 0.05; **p < 0.01, ***p < 0.001). **m** Representative immunofluorescence images of GFP-LC3 puncta (upper) and respective quantitative result in ENDOG or 14-3-3γ overexpressing L02 cells (lower). (HBSS: starvation for 6 hours; CQ, 50 μM for 6 h; scale bar = 10 μm; n = 100 independent cells examined over 3 independent experiments; data are presented as mean values ± SD,; ***p < 0.001). Source data are provided as a Source data file.

suggested that ENDOG-induced DNA damage may promote autophagy during starvation.

We next questioned if blocking DNA damage response can repress ENDOG-induced autophagy. Firstly, the comet assay showed that ENDOG indeed enhanced etoposide (DNA topoi-somerase II inhibitor) induced DNA damage (increased tail DNA and tail moment) (Fig. 6a–c). Moreover, treatment with etoposide not only enhanced the ENDOG-induced DNA damage response (increased p-H2A.X foci and p-ATM expression), but also increased autophagosome formation (GFP-LC3 puncta and LC3B-II accumulation) (Supplementary Fig. 10). At the same time, the loss of ENDOG significantly repressed the DNA damage response (Fig. 6d, e). Furthermore, etoposide-treated ENDOG knockout cells showed dramatically less p-H2A.X foci per cell, and the clearance of p-H2A.X foci in the recovery phase was significantly accelerated (Fig. 6f, g). These data demonstrated that ENDOG enhanced DNA damage response. To evaluate whether blocking DNA damage response pathway can sufficiently abolish ENDOG-induced autophagy, we performed experiments using an ATM specific inhibitor KU-60019. As expected, treatment with KU-60019 significantly reduced ENDOG-induced DNA damage response (p-ATM, p-CHK1, and p-CHK2 expression) and LC3B-II accumulation (Fig. 6h, i), as well as the number of p-H2A.X foci and autophagosome per cell (GFP-LC3 puncta) (Fig. 6j, k). However, ENDOG-overexpressing cells still had more p-H2A.X foci with the treatment of KU60019 (Fig. 6j, k), an inhibitor that almost completely abolished ATM activation but had no effect on ATR activation (Supplementary Fig. 11). These data suggest that ENDOG can also induce autophagy by activation of the DNA damage response.

**Endonuclease activity of ENDOG is essential for autophagy induction.** To examine which domain of ENDOG is essential for DNA damage and autophagy induction, we constructed various mutants, by deleting the mitochondrial targeting sequence (Del 1–48), mutating amino acids to eliminate its endonuclease activity (EM, enzyme mutation), and replacing mitochondrial targeting sequence with nuclear localization sequence (ENDOG-NLS) (Fig. 7a). We overexpressed these ENDOG mutants into the ENDOG knockout cells and treated them with etoposide. We found that the wild-type and NLS-ENDOG, but not the Del 1–48 and EM forms of ENDOG, had increased DNA damage in the ENDOG knockout cells (Supplementary Fig. 12a–c). The p-H2A. X foci staining also showed that the Del 1–48 and EM-ENDOG

had weaker DNA damage response after etoposide treatment (Supplementary Fig. 12d, e). Moreover, the expression of DNA damage sensor (PARP-1) and other DNA damage response proteins (p-ATM, p-ATR, p-CHK1, p-CHK2, and p-H2A.X) in Del 1–48 and EM-ENDOG groups were less than that in the wild-type and ENDOG-NLS groups (Supplementary Fig. 12f). However, only the EM-ENDOG failed to induced autophagy in the ENDOG knockout cells. Compared to the wild-type, Del 1–48 and ENDOG-NLS, the EM form of ENDOG overexpression has higher expression of p-mTOR, p-ULK1, p-p70S6K, and SQSTM1, suggesting EM form lose the ability to repress the mTOR activity and autophagy promotion (Fig. 7b, c). Moreover, overexpression of wild-type ENDOG and those mutants in ENDOG-KO cells revealed autophagic flux, except for the EM form (Fig. 7d, e). To further confirm that the DNA endonuclease activity is necessary for ENDOG-induced DNA damage and autophagy, we over-expressed the wild-type and EM-ENDOG in the ENDOG knockout cells. The comet assay results showed that EM-ENDOG could not induce DNA damage in the ENDOG knockout cell (Supplementary Fig. 13a–c). Furthermore, we found that com-pared to the wild-type ENDOG, the EM-ENDOG has less expression and activation of PARP-1, as well as less activation of AMPK and TSC2 both in the normal and starvation conditions (Supplementary Fig. 13d, e). Consistently, the mTOR activity in EM-ENDOG group is higher than that of the wild-type ENDOG group (Supplementary Fig. 13d, e), suggesting that EM-ENDOG lost the ability to repress mTOR. Furthermore, PNR-3-80, a specific ENDOG inhibitor[24], could repress the etoposide-induced DNA damage and autophagy (Supplementary Fig. 14). These results imply that endonuclease enzyme activity is essential for ENDOG-induced DNA damage response and autophagy.

Previous studies have demonstrated that caspase-8 cleaved and activated Bid, which mediates the translocation of ENDOG from mitochondria into the cytosol and ultimately into the nucleus, resulting in chromatin condensation and DNA fragmentation during apoptosis[12,25,26]. We, therefore, wondered if caspase-8/Bid is involved in our context. As expected, starvation caused the release of ENDOG from mitochondria into the cytosol and translocation into the nucleus (Supplementary Fig. 15a), and Z-IETD-FMK (a caspase-8 inhibitor) treatment efficiently decreased starvation-induced release of ENDOG from the mitochondria (cytochrome c staining) (Supplementary Fig. 15b–d). Meanwhile, activation of caspase-8/Bid and autophagy were consistently

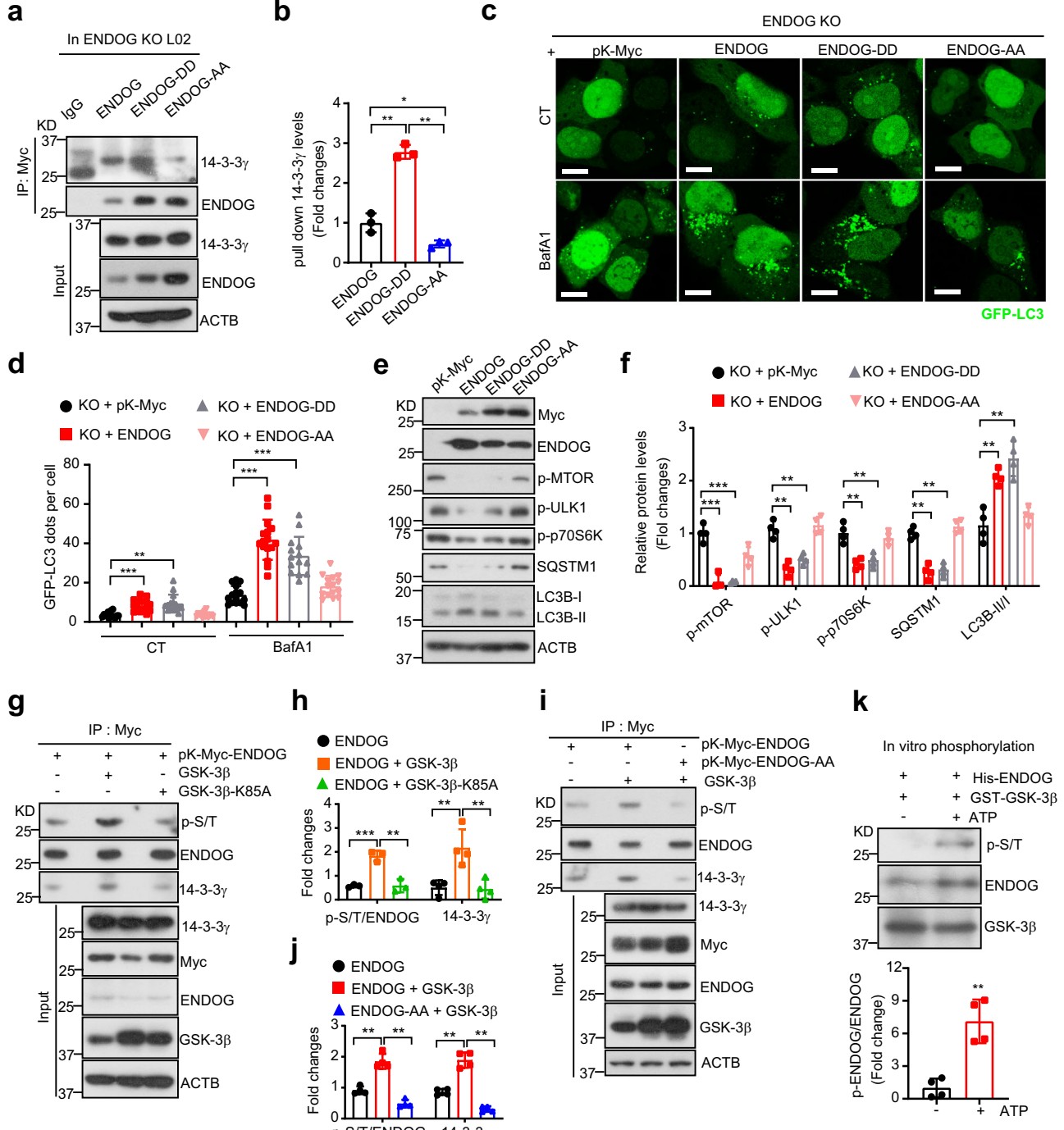

**Fig. 4 GSK-3β mediated phosphorylation of ENDOG at T128/S288 is necessary for ENDOG-induced autophagy. a, b** Co-IP experiments (**a**) and quantitative results (**b**) showed that phosphorylation of ENDOG at T128/S288 is necessary for its interaction with 14-3-3γ (overexpression of wild-type and mutant ENDOG in the ENDOG knockout L02 cell; WT: wild-type; DD: T128D/S288D; AA: T128A/S288A; $n = 3$ independent experiments; data are presented as mean values ± SD, *$p < 0.05$; **$p < 0.01$). **c, d** Representative immunofluorescence images of GFP-LC3 puncta (**c**) and respective quantitative results (**d**) (ENDOG knockout cells transiently transfected with the wild-type and mutant ENDOG for 48 h and treated with or without 100 nM BafA1 for 6 h; scale bar = 10 μm; $n = 100$ independent cells examined over three independent experiments; data are presented as mean values ± SD; **$p < 0.01$, ***$p < 0.001$). **e, f** Western blots (**e**) and quantitative results (**f**) of mTOR pathway and autophagy-related proteins. ENDOG-knockout cells were transiently transfected with the wild-type and mutant ENDOG for 48 h ($n = 4$ independent samples; data are presented as mean values ± SD; **$p < 0.01$, ***$p < 0.001$). **g, h**. Co-IP experiments (**g**) and quantitative results (**h**) showed that GSK-3β phosphorylates ENDOG and enhances the interaction between ENDOG and 14-3-3γ (transiently transfected of Myc-ENDOG, wild-type GSK-3β or kinase dead mutant GSK-3β-K85A in ENDOG knockout L02 cells for 48 h; $n = 3–4$ independent experiments; data are presented as mean values ± SD; **$p < 0.01$, ***$p < 0.001$). **i, j** Co-IP experiments (**i**) and quantitative results (**j**) showed that GSK-3β phosphorylates ENDOG at T128 and S288 (transiently transfected of Myc-ENDOG, Myc-ENDOG-AA or GSK-3β in ENDOG-knockout L02 cells for 48 h; $n = 4$ independent experiments; data are presented as mean values ± SD; **$p < 0.01$). **k** In vitro phosphorylation of ENDOG by GSK-3β by using the recombination protein (ATP: 5 μM; $n = 4$ independent experiments; data are presented as mean values ± SD, **$p < 0.01$). Source data are provided as a Source data file.

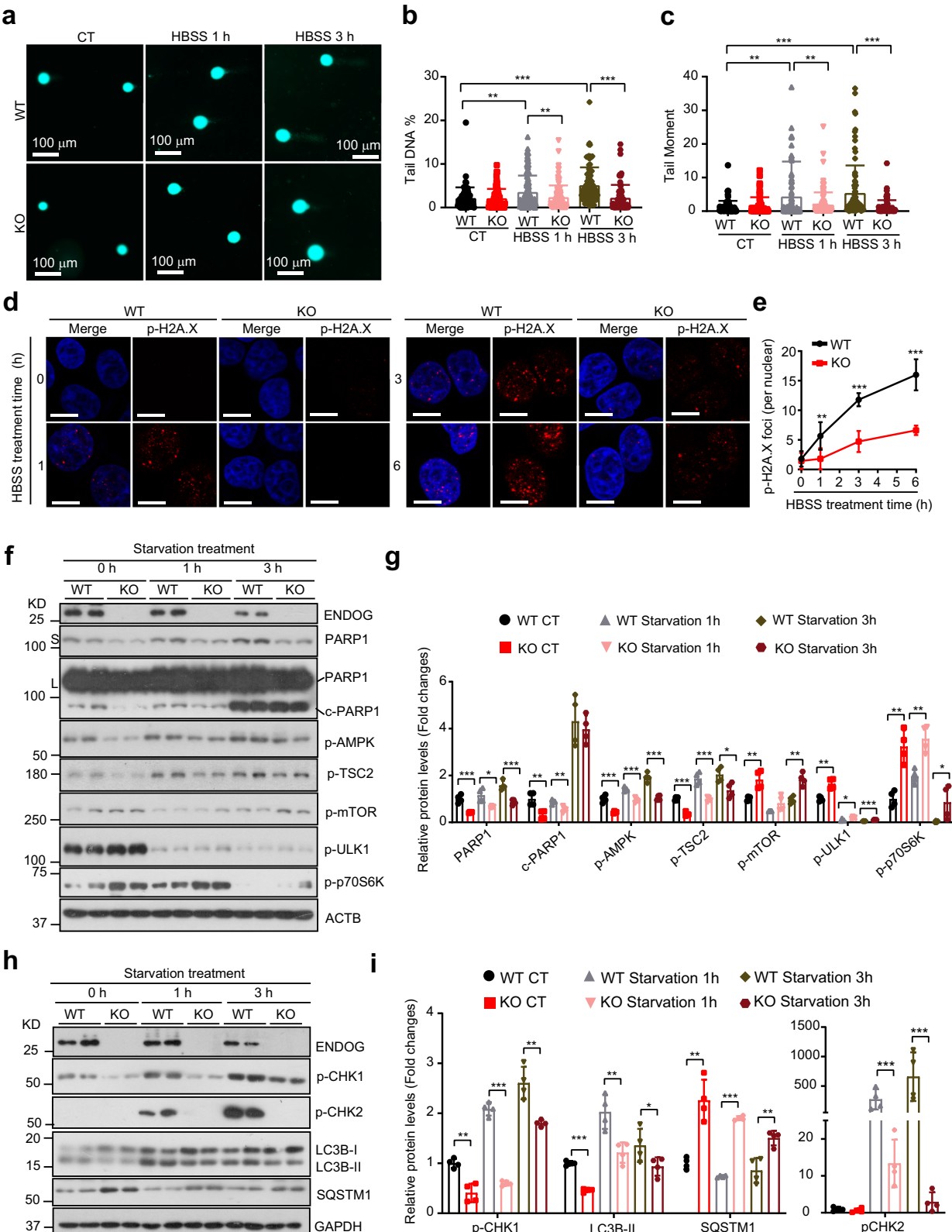

**Fig. 5 Loss of ENDOG repressed the starvation-induced DNA damage and autophagy. a–c** Representative images of comet assay in wild-type or ENDOG knockout L02 cells (**a**) and the quantification of tail DNA (**b**) and tail moment (**c**) (starvation treated for 1 or 3 h; $n = 75–100$ independent cells; data are presented as mean values ± SD, **$p < 0.01$, ***$p < 0.001$). **d, e** Representative images of p-H2A.X foci (**d**) and quantitative results (**e**) in wild-type or ENDOG knockout (KO) L02 cells at the indicated time point after the starvation treatment (scale bar = 10 μm, $n = 50$ independent cells; data are presented as mean values ± SD; **$p < 0.01$, ***$p < 0.001$). **f–i** Western blots and quantitative results of the indicated proteins in wild type or ENDOG knockout L02 cells following starvation treatment ($n = 4$ independent samples; data are presented as mean values ± SD, *$p < 0.05$; **$p < 0.01$; ***$p < 0.001$). Source data are provided as a Source data file.

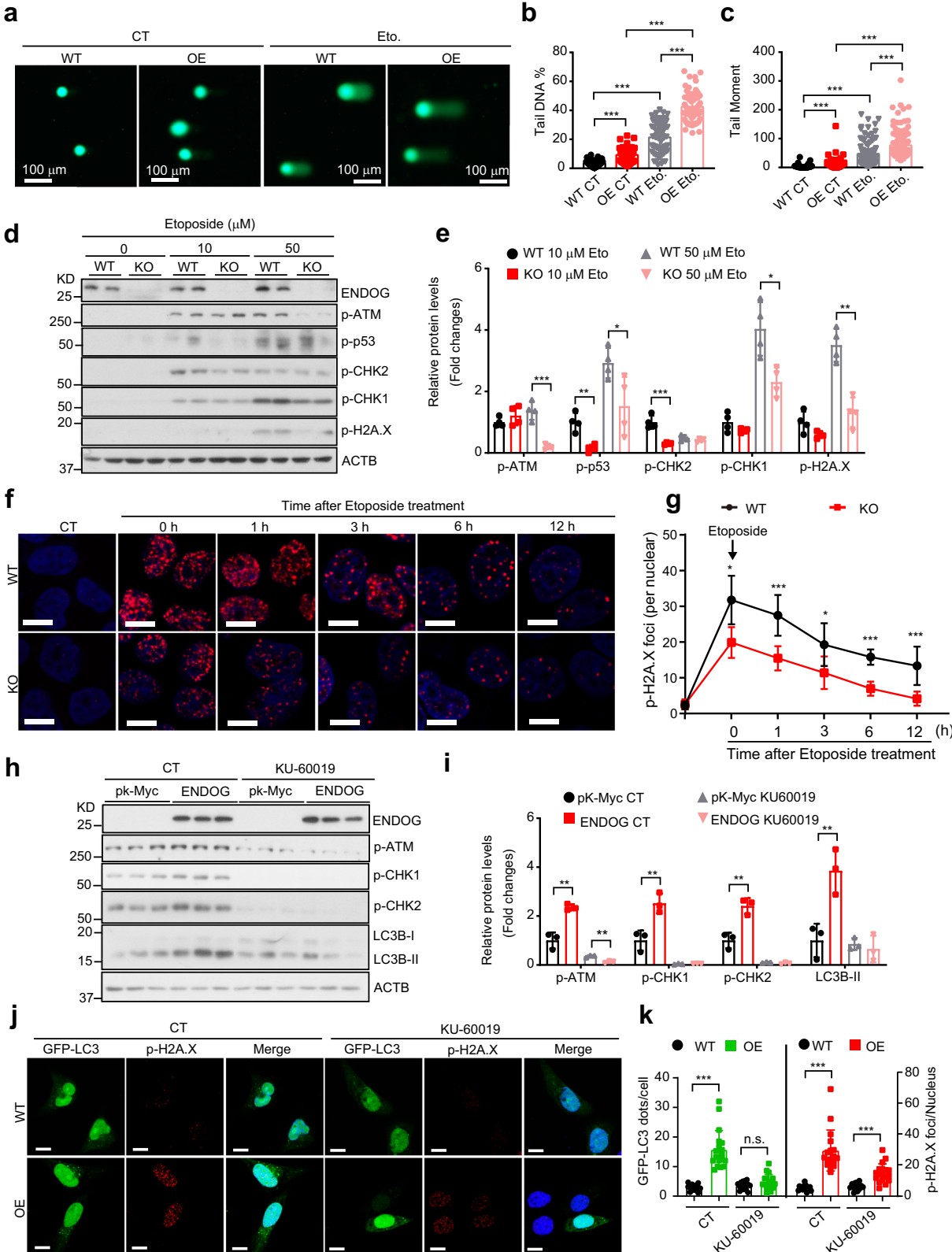

blocked by the caspase-8 inhibition (Supplementary Fig. 15e, f). These data suggest caspase-8/Bid contribute to ENDOG release from the mitochondria, which may further promote autophagy.

### Discussion

We reported an uncharacterized role of ENDOG to promote autophagy which is highly conserved in human cell lines, mouse,

*Drosophila* and *C. elegans*. Mechanistic studies showed that under stress conditions, caspase-8/Bid mediate the release of ENDOG from mitochondria. ENDOG is phosphorylated at T128 and S288 by GSK-3β, which enhances the interaction between ENDOG and 14-3-3γ, dissociates TSC2 and Vps34 from 14-3-3γ, and eventually suppresses the mTOR pathway and promotes autophagy initiation. In parallel, ENDOG can also translocate to the nucleus,

**Fig. 6 ENDOG promotes autophagy by activation of the DNA damage response. a–c** Representative images of comet assay in wild-type or ENDOG overexpressed L02 cells (**a**) and the quantification of tail DNA (**b**) and tail moment (**c**) (WT: wild-type; OE: ENDOG-overexpressing; Eto.: 50 μM etoposide for 1 h; $n = 75$–150 independent cells; data are presented as mean values ± SD, ***$p < 0.001$). **d, e**. Representative western blot results of phosphorylated ATM, p53, CHK1, CHK2 and H2A.X in wild-type or ENDOG knockout (KO) L02 cells following the indicated concentration of etoposide treatment for 1 h ($n = 4$ independent samples; data are presented as mean values ± SD, *$p < 0.05$; **$p < 0.01$, ***$p < 0.001$). **f, g** Representative images of p-H2A.X foci (**f**) and quantitative results (**g**) in wild-type or ENDOG knockout (KO) L02 cells at the indicated time point after the etoposide treatment (scale bar = 10 μm, $n = 50$ independent cells; data are presented as mean values ± SD, *$p < 0.05$; ***$p < 0.001$). **h, i** Western blots (**h**) and quantitative results (**i**) of phosphorylated ATM, CHK1, CHK2, and LC3B upon ATM inhibitor (KU-60019) or control treatment (L02 cells were transfected with pK-Myc or ENDOG or 48 h and then treated with 10 μM KU-60019 for 1 h; $n = 3$ independent samples; data are presented as mean values ± SD, **$p < 0.01$). **j, k** Representative images and respective quantitative results of p-H2A.X foci and GFP-LC3 puncta in wild-type or ENDOG-OE cells upon ATM inhibitor (KU-60019) or control treatment (scale bar = 10 μm; $n = 100$ independent cells examined over three independent experiments; data are presented as mean values ± SD; ***$p < 0.001$; n.s.: no significance). Source data are provided as a Source data file.

activate the DNA damage response pathway and enhance autophagy via its endonuclease activity (Fig. 7f).

As phosphorylation readers, 14-3-3 proteins regulate the function of their target proteins by binding to their phospho-serine or phosphothreonine motifs[20]. 14-3-3 proteins have been shown to regulate autophagy through binding with the proteins that regulate the mTOR pathway (TSC2, PRAS40 and Raptor) or autophagy initiation (Beclin-1, ULK1, and Vps34)[7–11]. 14-3-3 also forms a complex with phosphorylated Beclin-1 and vimentin to promote tumorigenesis by inhibiting autophagy[27,28]. PI3K inhibitors (BKM120) promote the dissociation of FOXO3a from 14-3-3ζ to induce autophagy[29]. The GSK3 inhibitor induces TFEB dephosphorylation and its dissociation from 14-3-3, resulting in TFEB activation and nuclear translocation to activate autophagy[30]. All these studies support that 14-3-3 proteins play crucial roles in autophagy by interaction with different proteins involved in autophagy. Consistently, in our study, we revealed that ENDOG binding with 14-3-3γ depends on the phosphor-ylation of ENDOG at Thr-128 and Ser-288 (Fig. 4a). ENDOG competitively interacts with 14-3-3γ and promotes the release of TSC2 and Vps34 from 14-3-3γ, which suppress mTOR signaling and promote autophagy initiation (Fig. 3i–j).

Under stress stimuli, caspase-8 is activated and cleaves Bid into tBid, which mediates the release of AIF and ENDOG from mitochondria[12,25,31]. Our study successfully confirmed that (Supplementary Fig. 14a, e), and we found caspase-8 inhibition represses the release of ENDOG and starvation-induced autop-hagy (Supplementary Fig. 14c–e). We identified that GSK-3β mediated the phosphorylation of ENDOG (Fig. 4g–k). Since AKT phosphorylates GSK-3β at Ser9 which inactivates GSK-3β[32], overexpression of an activated form of AKT significantly repressed the phosphorylation of ENDOG (Supplementary Fig. 9). However, whether this phosphorylation participates in the nuclear translocation of ENDOG is not yet known.

ENDOG is a DNase that generates DNA fragmentation during apoptosis[12]. It is capable of digesting double or single-strand DNA and DNA-RNA heteroduplexes[13]. Single-strand DNA breaks in $(dG)_n\cdot(dC)_n$ are generated preferentially when ENDOG fragments double-stranded DNA[33]. We suspected that ENDOG nuclease activity might induce the DNA damage response pathway. We demonstrated that ENDOG promoted DNA damage both under normal and stress conditions (Supplementary Fig. 10–13). DNA damage has been reported as an early event during starvation-induced autophagy. Under starvation, PARP-1/AMPK and ATM/CHK2 pathways were activated and eventually promoted autophagy[21,22]. In the present study, ENDOG promoted starvation-induced DNA damage through PARP-1/AMPK pathway, which could repress the mTOR activity and initiate autophagy (Fig. 5). In contrast to the wild-type ENDOG, the endonuclease activity defi-cient form of ENDOG could not activate the PARP-1/AMPK pathway or induce mTOR repression and autophagy promotion

(Supplementary Fig. 13d, e). In addition, recent study showed that DNA damage induced CHK2-mediated FOXK phosphorylation, which traped FOXK in the cytoplasm through interacting with 14-3-3, resulting in promoting the transcriptional of autophagy-related genes (ATGs) and facilitating autophagy[21]. However, the tran-scription of autophagy-related genes was not affected in ENDOG overexpression or knockout cells (Supplementary Fig. 1c, d). This suggested that ENDOG-mediated DNA damage response promotes autophagy in a CHK2-FOXK-axis independent manner.

Previous studies showed that DNA damage agents (such as camptothecin, etoposide, and temozolomide)[34,35] and ionizing radiation[36] promoted autophagy initiation. Consistently, we found etoposide treatment promoted autophagosome formation (GFP-LC3 dots) both in wild-type and ENDOG-overexpressing cells (Supplementary Fig. 10). Moreover, by using the ATM inhibitor KU-60019, which only inhibits ATM but not ATR, to block the DNA damage response, ENDOG-induced DNA damage and autophagosome formation could be partially repressed (Fig. 6h–k and Supplementary Figs. 10, 11).

Additionally, ENDOG without endonuclease activity did not induce the DNA damage response or autophagy (Supplementary Figs. 12, 13). Chemical inhibition of ENDOG activity also repressed DNA damage and autophagy (Supplementary Fig. 14). These data suggest that ENDOG-promoted autophagy is endo-nuclease activity-dependent.

ENDOG's N-terminus (1–48 aa) is a mitochondrial targeting sequence (MTS) that is removed after ENDOG is imported into the mitochondria[13,37]. Interestingly, we found that ENDOG without MTS failed to induce DNA damage response, but it could still promote autophagy (Supplementary Fig. 12). Therefore, we propose that ENDOG without MTS lost its ability to cause DNA damage in nuclei, because it did not mature properly without being imported into mitochondria[13] or because it was affected in another unknown way. However, ENDOG without MTS main-tained its binding affinity with 14-3-3γ, so it can still repress mTOR pathway and promote autophagy initiation. This may be the reason why Del 1–48-ENDOG did not induce DNA damage but still promoted autophagy under the etoposide treatment (Supplementary Fig. 12).

In summary, our findings indicate that ENDOG conservatively promotes autophagy in multiple species. In mammalian cells, ENDOG promotes autophagy through the suppression of mTOR by its phosphorylation-mediated interaction with 14-3-3γ and its endonuclease activity-mediated DNA damage response.

## Methods

**Animals**. The ENDOG-knockout mice (C57BL/6) were generated by Cyagen Biosciences (Guangzhou) Inc. using the CRISPR/Cas9 system. Briefly, gRNA to mouse *Endog* (gRNA1 (matching the reverse strand of the gene): AGCGCGCG CATGCCTACCGGAGG, gRNA2 (matching the reverse strand of the gene): AGCTCTAAAGGCACGGGGACTGG) and Cas9 mRNA were co-injected into fertilized mouse eggs to generate targeted knockout offspring. F0 founder animals

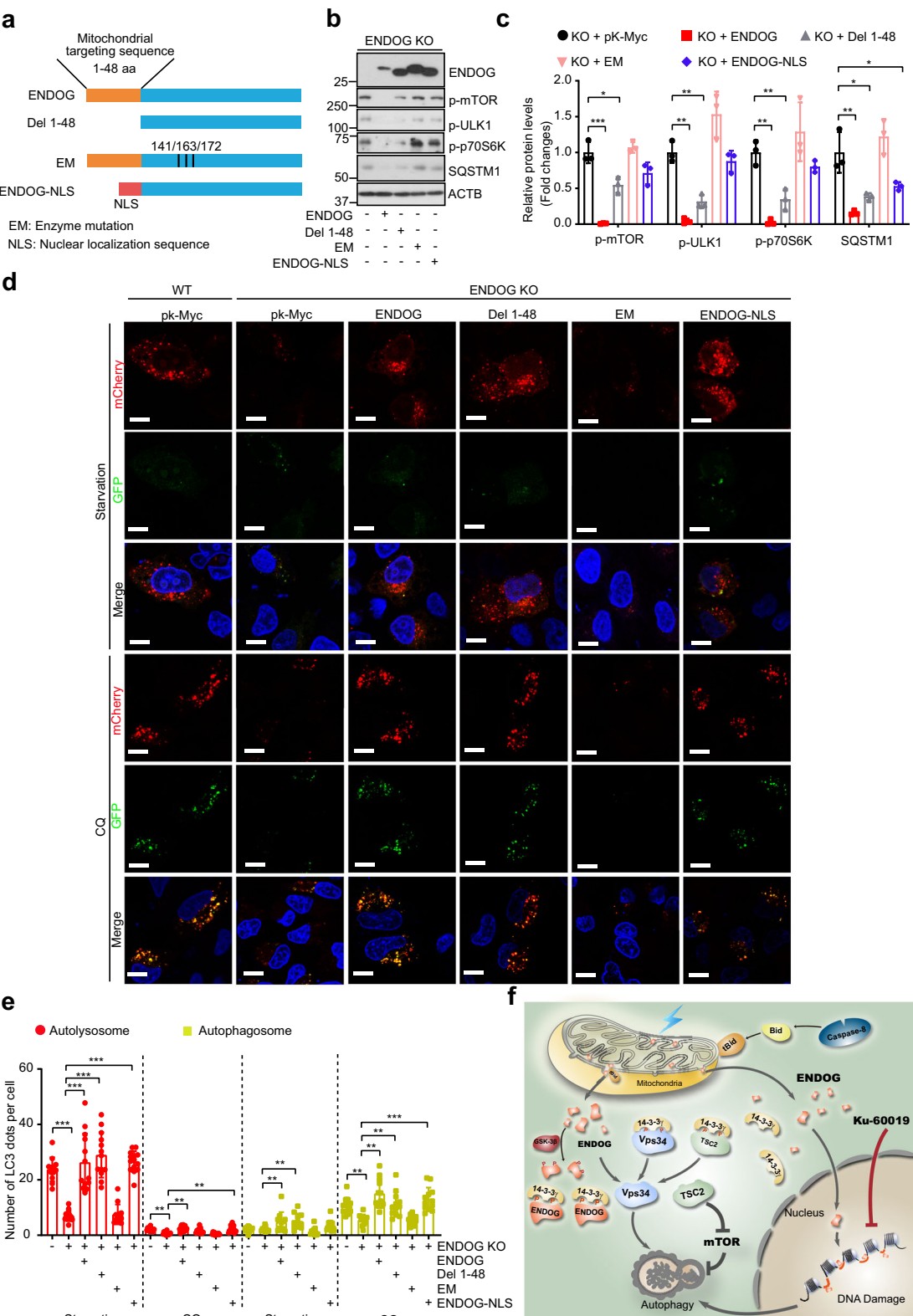

were identified by PCR followed by sequence analysis. F0 mice were mated with the wild-type mice (C57BL/6) to generate the F1 mice (Endog$^{+/-}$); F1 male mice were crossed with F1 female mice to generate the F2 mice (Endog$^{+/-}$ and Endog$^{-/-}$) for experiments. Three-month-old male mice (Endog$^{+/-}$ and Endog$^{-/-}$) were used in this study. Mice were maintained on a 12 light/12 dark cycle (light on 7 AM to 7 PM), at a room temperature of 22 °C ± 2 °C, humidity of 50% ± 5%, with free access to food and water. The animals were handled according to the Guidelines of the China Animal Welfare Legislation and as approved by the Laboratory Animal Ethics Committee of JINAN University.

**Cell culture and treatment**. Human fetal hepatocyte cells (L02) and hepatoma cells were maintained in DMEM (Gibico, C11995500BT) supplemented with 10% fetal bovine serum (HyClone, SV30160.03), 1% penicillin and streptomycin (Gibico,15140-122). All cells were incubated at 37 °C in 5% CO$_2$. For starvation, the cells were cultured in serum-free Hanks balanced salt solution (HBSS) medium (HyClone, SH30030.02) for 6 or 12 hours, treated with rapamycin (1 μg/ml, Med Chem Express, 53123-88-9) or chloroquine (CQ, 50 μM, Sigma, C6628) for 6 h. For DNA damage treatment, the cells were exposed with etoposide (50 μM, Sigma, E1383) for 1 h or the ATM inhibitor KU-60019 (10 μM, Selleckchem, S1570) for

**Fig. 7 Endonuclease activity of ENDOG is essential for autophagy induction. a** Schematic diagram of human wild-type ENDOG and ENDOG mutations. Del-1–48, deletion of the N-terminal 48 amino acids; EM, endonuclease activity mutation by mutagenized H-N-N motif (141H/163 N/172N) to 141A/163 A/172A; ENDOG-NLS, replacement of the mitochondrial targeting sequence with a nuclear localization signal sequence. **b**, **c** Western blots (**b**) and quantitative results (**c**) of the indicated proteins in ENDOG knockout L02 cells following overexpression of wild-type or mutant ENDOG and starved for 6 h ($n = 3$ independent samples; data are presented as mean values ± SD, $*p < 0.05$; $**p < 0.01$, $***p < 0.001$). **d**, **e** Representative images (**d**) and respective quantitative results (**e**) of mCherry-GFP-LC3 puncta in the indicated cell groups under starvation and CQ treatments (WT: wild-type L02 cells; KO: ENDOG knockout L02 cells; plasmids transiently transfected for 48 h; starvation for 6 h; CQ: 50 µM for 6 h; scale bar = 10 µm; $n = 100$ independent cells examined over three independent experiments; data are presented as mean values ± SD, $**p < 0.01$, $***p < 0.001$). **f** Model depicting ENDOG promotes autophagy through its phosphorylation-mediated interaction with 14-3-3γ and its endonuclease activity-mediated DNA damage response. Source data are provided as a Source data file.

the indicated experiments. To inhibit caspase-8 activity, cells are exposed with Z-IETD-FMK (50 µM, Selleckchem, S7314 for 2 h). For inhibition of the ENDOG enzyme activity, cells were treated with the ENDOG specific inhibitor PNR-3-80 (provided by Dr. Alexei G. Basnakian) for 24 h at the concentration of 50 µM.

**Plasmids, virus production, and transfection**. The open reading frames (ORFs) of ENDOG, 14-3-3γ and RHEB/RHEB$^{Q64L}$ were amplified by PCR from cDNAs, then cloned into the pK-Myc or pcDNA3.1 vector. Mutant forms of ENDOG, i.e. mitochondrial sequence deletion (Del 1–48), endonuclease activity null (EM) and nuclear localization sequence addition (NLS) forms, were generated using a QuickChange site-directed mutagenesis kit (Agilent Technologies, 200518). To generate the stable ENDOG-overexpressing cell line, the ENDOG ORF were cloned into the lentiviral vector pLvx3. To produce lentivirus, these plasmids were transfected into 293T cells together with packaging plasmids (psPAX2, pMD2.G, gifts from Dr. Zhiyin Song, Wuhan University). Lentivirus particles were collected and used to infect the target cells. Stable ENDOG-overexpressing cell lines were selected with 1µg/ml puromycin (Invitrogen, A1113802). GSK-3β and GSK-3β-K85A were purchased from Addgene (Catalog: 14753 and 14755). All the plasmids used for the indicated experiments were transfected using Lipofectamine LTX (Invitrogen, 11668-019).

**Generation of ENDOG knockout cell lines**. We generated ENDOG knockout cell lines using the CRISPR/Cas9 system. Two different single guide RNA (sgRNA) sequences targeting ENDOG were obtained from a free online CRISPR design tool (crispr.mit.edu). The two sgRNA sequences are sgRNA1: CCGGGCGAGCTGGC CAAGTA and sgRNA2: CGACTTCCGCGAGGACGACT. Annealed double-stranded sgRNA oligos were ligated into the lentiCRISPR vector pRX00-1, which co-expresses Cas9 and sgRNA in the same vector. Cells transfected with the pRX-001-sgRNA-ENDOG. Positive clones were selected with 1µg/ml puromycin, PCR genotyping and sequencing was used to choose the ENDOG knockout clones.

**Western blot**. Western blot analyses were performed as previously described[38]. Briefly, samples were lysed with RIPA buffer (Beyotime, P0013B) with protease inhibitors and sonicated. Then the protein concentrations were measured with a Pierce BCA protein assay kit (Thermo Scientific, 23225) and loaded onto SDS-PAGE gels for western blot analysis. The primary and secondary antibodies used are shown in Supplementary Table 2. Protein bands were evaluated using Quantity One 1-D analysis software (Bio-Rad, Hercules, CA). Protein levels were quantified relative to ACTB/β-actin in the same sample, and the relative protein expression was normalized to the respective control group, which was set to 1.

**Quantitative real-time PCR (qRT-PCR)**. Total RNA from the cells or tissues was isolated using RNAiso Plus reagent (TaKaRa Biotechnology, 9109) following the manufacturer's protocol. One microgram of total RNA was reverse transcribed into cDNA using the ABScript II cDNA First Strand Synthesis Kit (ABclonal, RK20400) following the manufacturer's protocol. Then the mRNA levels were quantified with a SYBR Green Select Master Mix (ABclonal, RK21203) on a CFX96 real-time system (Bio-Rad). Relative mRNAs levels were calculated using the $2^{-\Delta\Delta CT}$ method, with Actb used as the internal control. Primer sequences for the target genes are provided in Supplementary Table 3.

**Quantification of GFP-LC3 puncta**. Cells (L02 or HepG2) were transfected with the GFP-LC3 plasmid for 48 h. After the indicated treatment, cells were fixed with 4% paraformaldehyde and imaged with a Leica TCS SP8 confocal microscope. GFP-LC3 puncta in at least 100 cells were counted for per treatment. A diffused distribution of GFP-LC3 indicated non-autophagic puncta.

**mCherry-GFP-LC3 reporter assay**. mCherry-GFP-LC3 reporter plasmid which expressed LC3 fused with mCherry and GFP, were used to detect the autophagic flux. The green fluorescent signal of GFP is sensitive to the acidic environment of lysosomes, whereas the red fluorescent signal of mCherry is more stable. Therefore, colocalization of GFP and mCherry fluorescence (yellow puncta) indicated an

autophagosome, and detection of mCherry but not GFP (red puncta) indicated an autolysosome. Cells were transfected with the mCherry-GFP-LC3 reporter plasmids for 48 h. After the treatment, cells were fixed and imaged. Yellow and red puncta were counted for each cell line. At least 100 cells were analyzed per treatment condition.

**Electron microscopy**. ENDOG knockout and wild-type L02 cells were grown in 10-cm dishes and treated with 100 nM BafA1 for 6 h. Three-month-old Endog$^{+/-}$ and Endog$^{-/-}$ mice were starved for 12 h. After the treatment, cells or liver tissues were collected for the electron microscopy assay. Cells or the fresh liver tissue were fixed in 2.5% glutaraldehyde for 12 h at 4 °C. After fixation, cell monolayers were washed three times in PBS and then post-fixed with 1% osmium tetroxide in 0.1 M PBS for 2 h at room temperature. After triple wash with PBS, the samples were dehydrated through a graded series of ethanol and embedded in EMBed 812 epoxy resin (SPI, 90529-77-4). Ultrathin (60 nm) sections were collected on copper grids and stained with 2% uranyl acetate in saturated alcohol for 8 min, followed by 2.6% lead citrate exposure for 8 min. Images were photographed with a Hitachi HT7800 transmission electron microscope. The electron microscopy assays were done in cooperation with Servicebio (Wuhan, China).

**Co-immunoprecipitation and mass spectrometry analyse**. Cells were transfected with different plasmids for 48 h and then lysed with Western/IP buffer with protease inhibitors (Beyotime, P0013J) and sonicated. Protein concentrations were measured with a Pierce BCA protein assay kit (Thermo Scientific, 23225). Myc (Proteintech, 16286-1-AP), 14-3-3γ (Proteintech, 12381-1-AP) or Flag (Sigma, F1804) antibodies were used to immunoprecipitate tagged 14-3-3γ, endogenous 14-3-3γ, ENDOG or ENDOG mutant forms. The precipitates were boiled and loaded onto SDS-PAGE gels for western blot analysis with secondary antibodies (Mouse Anti-Rabbit IgG (Light-Chain Specific) (D4W3E) #93702 or VeriBlot for IP Detection Reagent (ab131366). For mass spectrometry analysis, specific protein compounds were excised and, following trypsin digestion, were subjected to HPLC-MS analysis. Data analysis was performed by Fitgene Biotech Co. (China, Guangzhou). The identified proteins were verified by co-IP experiments.

**Autophagy analysis in mice, _C. elegans_ and _Drosophila_**. Three-month-old male (C57BL/6) ENDOG knockout (Endog$^{-/-}$) and control mice (Endog$^{+/-}$) were fasted for 24 h, and autophagy levels in mice livers were detected by western blot. The ENDOG mutant _C. elegans_, _cps-6_ (_tm3222_) was hybridized with the GFP:: LGG-1 strain. PCR genotyping was used to verify the _cps-6_; GFP::LGG-1 strains. For autophagy analysis, the GFP::LGG-1 and _cps-6_; GFP::LGG-1 strains were starved for 4 h. Following the starvation treatment, GFP::LGG-1 puncta in seam cells and pharyngeal muscle were imaged and counted. Autophagy analysis in _Drosophila_ second-instarlarvae were collected 72–96 h after the eggs were laid and cultured in vials containing 20% sucrose (starvation conditions) for 4 h. Autophagy levels were determined by the number of mCherry-ATG8a-labeled autophagosomes. GFP-marked RNAi clones (RNAi lines are BDSC55228 (#1) and BDSC62903 (#2)) in the larval fat body were generated by heat shock-independent induction as previously described[39].

**Comet assay**. The Comet assays were performed using COMET Assay kit (Enzo, ADI-900-166) following the manufacturer's instructions. $1 \times 10^5$ cells/ml were mixed with molten LM agarose at 37 °C at a ratio of 1:10 (vol/vol) and pipetted onto a COMET slide. The slides were placed for 30 min in the dark at 4 °C and were immersed in pre-chilled lysis solution. The slides were then removed from lysis buffer, washed in TBE buffer and transferred to a horizontal electrophoresis chamber. Voltage (1 V/cm) was applied for 20 min. The slides were washed in distilled water immersed in 70% ethanol for 5 min and allowed to air dry. Slides were stained with SYBR Green and then analyzed by fluorescence microscopy. 70–150 cells were evaluated in each sample using the COMET Assay Software Project (CASP software). Tail DNA% = Tail DNA/ (Tail DNA + Head DNA), Tail moment = Tail length × Tail DNA%.

**Subcellular fractionation.** Cell fractionation and mitochondria isolation were performed as previously described[40]. Briefly, following the 12 h starvation treatment, cells were collected and centrifuged at $400 \times g$ for 5 min (at 4 °C). Cell pellet was permeabilized with 10 mg/ml digitonin (Sigma, D141) in PBS for 10 min at room temperature. Cells were centrifuged at $10,000 \times g$ for 10 min (at 4 °C), and supernatants were collected to a fresh tube (cytosolic fraction). The pellets were resuspend in the mitochondrial lysate buffer (10 mM Tris-HCl pH 7.4, 150 mM NaCl, 2 mM EDTA, 0.2% Triton, 0.3% NP40, protease inhibitor cocktails) for 30 min on ice and centrifuged at $10,000 \times g$ for 20 min (at 4 °C). The supernatants were the mitochondrial fraction and the pellets were nuclear fraction.

**Immunofluorescence staining.** For immunofluorescence staining, the cells were fixed with 4% formaldehyde and blocked with 2% BSA (Amresco, E588), then incubated with anti-p-H2A.X (Cell Signaling Technology 9718, 1:500 dilution), Cytochrome c (Proteintech, 66264-1-Ig, 1:500 dilution) or ENDOG (NOVUS, IMG-5565-2, 1:100 dilution) overnight at 4 °C. The cells were then incubated with Alexa Fluor® 594-AffiniPure goat anti-rabbit (Jackson, 115-585-146) or Alexa Fluor® 488-AffiniPure goat anti-mouse IgG (Jackson, 115-545-146) and counterstained with DAPI. Images were taken with a Leica TCS SP8 confocal microscope.

**p-H2A.X foci assays.** The p-H2A.X foci assays were performed as in a previous study[41]. Briefly, wild-type and ENDOG knockout L02 cells were seeded in the 12-well plate and incubated for 24 h. After treatment with 50 μM etoposide for 1 h, cells were washed with warm PBS twice and transferred to fresh medium. Then cells were fixed at different time points for plotting the kinetics of p-H2A.X induction and clearance. The fixation was in ice cold 4% paraformaldehyde for 10 min at room temperature, cells were then permeabilized with 0.1% Triton X-100:PBS, blocked in 5% BSA, and stained with anti-p-H2AX antibody (CST, 9718) overnight and anti-RabbitAlexaFluor-488 secondary antibody (Jackson, 115-585-146) for 2 h at room temperature. DAPI was used to counterstain nuclei. Images were taken with a Leica TCS SP8 confocal microscope. The p-H2A.X foci were scored manually using Image-Pro-Plus 6.0 software, and the average number of foci per cell was calculated from a minimum of 100 cells per time point. Experimental data represent the average of two independent experiments.

**In vitro phosphorylation.** The in vitro phosphorylation assay was performed as described previously[42] with modifications. Briefly, purified GST-GSK3β (Proteintech, Ag3688) and His-ENDOG (Proteintech, Ag17847) were incubated for 1 h at 37 °C in kinase buffer (50 mM Tris (pH 8.0), 10 mM MgCl₂, 1 mM DTT) with or without 5 μM ATP. After the reaction, the mixtures were added 5× SDS loading buffer and boiled at 95 °C for 15 min and then separated on SDS-PAGE gels. Phosphorylation of ENDOG was detected by the Phospho-Ser/Thr antibody (CST, 25081).

**Statistics and reproducibility.** All the experiments were performed at least three independent experiments or sufficient sample sizes were involved. All statistical analyses were performed using GraphPad Prism software. All statistical results are presented as the mean ± standard deviation (S.D.). Difference between three or more means was assessed using One-way ANOVA, difference between two means was assessed by using the two-tailed, unpaired Student's $t$-test in GraphPad Prism version 8.0; $p$ values < 0.05 were considered statistically significant.

**Reporting summary.** Further information on research design is available in the Nature Research Reporting Summary linked to this article.

## Data availability
All relevant data are available from the corresponding author upon any reasonable request. Source data are provided with this paper.

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

## Acknowledgements
We thank Dr. Guang Yang for the mCherry-GFP-LC3 and GFP-LC3 plasmids, Dr. Laixin Xia for the CRISPER/Cas9 system, Dr. Liang Chen for the HepG2, MHCC97-H and PLC/PRF/5 cells, Dr. Xuhui Lai for technical help on qPCR procedure, Dr. Stephanie Mohr for proofreading. This work was financially supported by the National Natural Science Foundation of China (Grants. 81800833 and 81802189), the 111 Project (B16021), and the National Key R&D Program of China (2018YFC2002000), the China Postdoctoral Science Foundation (grant 2018M631054), and the Natural Science Foundation of Guangdong Province (grant2018A0303131002, 2019A1515011847, and 2019A1515010591). Work in N.P.'s laboratory was supported by P01CA120964, the STARR foundation and HHMI. ENDOG inhibitor synthesis and testing in A.B.'s laboratory was supported by 2P20GM109005 and 2I01BX002425.

## Author contributions
The project was supervised by Q.Z. Experiments were designed by J.L. and Q.Z., performed by W.W., J.L., J.T., M.W., J.Y., and H.T., analyzed by W.W. and J.L. Z.Z., C.L., H.T., and N.P. provided technical support and discussions, A.B. provided the ENDOG inhibitor and discussions. W.W., J.L., and Q.Z. wrote the manuscript. All the authors reviewed the manuscript.

## Competing interests
The authors declare no competing interests.
