## [Peer Review File · Nature Communications]

Reviewers' comments:

Reviewer #1 (Remarks to the Author):

General Comments:

This manuscript, entitled "Endonuclease G promotes autophagy by suppressing mTOR signaling and activating the DNA damage response", investigated the function and underlying mechanisms of ENDOG in regulating autophagy. Authors tried to conclude that ENDOG promotes autophagy during starvation by phosphorylation-mediated interaction with 14-3-3g, and its endonuclease activity-mediated DNA damage response. The major concern of this manuscript is lack of novelty. EndoG functions as a paternal mitochondria degradation factor through autophagy machinery, which has been reported in 2016. All the mTOR and autophagy pathways studied in this manuscript have been well established/predicted in the field already. Authors mainly validate the pathways in this manuscript. The rationale of this study is not strong. Moreover, the evidence for the key conclusion that endoG promotes autophagy is not convincing. This study also heavily relies on the artifact overexpression system. Many important conclusions were made based on the artifact overexpression system. Lots of other concerns are shown as following.

Major Comments:

1. Authors concluded that endoG promotes autophagic flux mainly based on the increase of LC3B-II in endoG over-expression cells. However, LC3B-II is not a specific marker for autophagy, as recent studies showed its role in phagocytosis (Cunha LD, et al., Cell 2018) and endocytosis (Heckmann BL et al., Cell 2019). Additional specific autophagy markers should be tested in order to support authors' conclusion.
2. Fig. 1A, why the expression of LC3B-I and LC3B-II was much higher in endoG-overexpressing cells than pk-Myc cells? Why there was no expected reduction on LC3B-I after it was converted into LC3B-II? Authors should exclude the possibility that was caused by the unequal loading.
3. Fig. 1B, it is not clear what and why is the strong green signal in the nucleus on the left panel (pk-Myc). There is no sufficient data explanation. The results of this part as well as many other parts were not clearly described.
4. Fig. 1E, why two repeats have so different results-SQSTM1?
5. Fig. 1F, authors compared the protein levels in CT and BafA1 groups on two separated blots. These two groups have to be run on the same blot in order to do such comparison.
6. Fig. S1A, no reduction was observed with SQSTM1. Again, why overall expression of LC3B-I and II was increased? Why there was no expected reduction of LC3B-I, when it was converted into LC3B-II?
7. EndoG knockout mice information was not provided in this manuscript. It was not clear how the knockout mice were made and how old the mice were used in the experiments.
8. Line 111, authors claimed "showed decrease LC3B accumulation". However, no obvious difference was observed in Fig. 2A to support authors' conclusion.
9. Authors concluded that 14-3-3y interacts with endoG, which was not convincing. It was only observed in an artifact overexpression system. Authors also claimed that Starvation enhanced their interaction, which was also not true. No specific interaction signal was observed in Fig. 3J.
10. The link between DNA damage response and endoG-mediated autophagy was rather weak. It was not clear what was the rationale to use etoposide to damage DNA and how it could be linked to starvation-induced autophagy. It was not a surprise to see the involvement of endoG in DNA damage response according to previous studies. However, this study did not show how endoG was involved in DNA damage response. Authors claimed that endoG mediated DNA damage response by its endonuclease activity. However, there is no any direct evidence to support this conclusion. It is mandatory to use its endonuclease deficient mutant to confirm their conclusion.
11. Fig. S9, p-H2Ax should be checked at multiple time points. 1 h after etoposide is a little bit too short usually.

Minor Comments:

1. Grammar errors. For example, "A previous results..." on page 4, line 63.
2. Fig. 1, it is better to label each panel, including quantification panels.
3. Fig. S2A, what is the difference between the left and right panels? The label is missing.

Reviewer #2 (Remarks to the Author):

This manuscript reported that the Endonuclease G (ENDOG) promotes autophagy through inhibiting mTOR and activating DDR. Mechanistically, ENDOG is phosphorylated by GSK-3 β , which enhance its interaction with 14-3-3, leads to the release of TSC2 and Vps34, and inhibit mTOR. The author also showed that ENDOG promotes DDR initiation. However, the molecular mechanism is still not clear. For example, how ENDOG activate DDR? How EM mutant affect mTOR signaling? And some data is not in high quality. There are currently a number of shortcomings with the experimentation I listed below that should be addressed in order to strengthen the findings.

1. In Fig 3I,J, for the Co-IP data, the blot for TSC2(I) and ENDOG(J) is in low quality. Please repeat this experiment and show better blot data.
2. In Fig 3K, The LC3B level of ENDOG+14-3-3 group is higher than 14-3-3 group. But in Fig 3M, they are similar. The author should explain the inconsistent results.
3. In Fig 4, due to mTOR activation, The LC3B level of ENDOG-AA should be decreased compared to ENDOG-WT and DD groups. But the blot result is opposite and it is not match with the quantification data. The author should explain that and check the data more carefully.
4. In Fig 4J, the label is wrong. That should be ENDOG-AA but not DD.
5. In Fig 5G-H, with ATM inhibition, overexpression of ENDOG cannot increase LC3 punta. However, overexpression of ENDOG should still activate autophagy through competing 14-3-3 binding. Does that mean ATM also affect GSK-3 β signaling? The author should perform further experiment to confirm the result. On the other hand, overexpression of ENDOG increased γ -H2AX foci with ATM inhibition. What is the mechanism?
6. From Fig 5, it looks like that ENDOG promotes DDR initiation, such as p-ATM and γ -H2AX at early time point. What is the mechanism? At least, the author should test the different ENDOG mutants'(listed in Fig 6A) function in DDR activation.
7. In Fig 6, the author suggested that EM mutant cannot rescue LC3B punta because of its activation of mTOR. My question is how EM mutant (endonuclease activity deficient) affect its binding with 14-3-3 and activate mTOR? Does it affect GSK3- β phosphorylation on ENDOG? Also, does EM mutant affect DDR? All these are important question to clarify the mechanism that how ENDOG activate autophagy.

Reviewer #3 (Remarks to the Author):

In the manuscript, the authors studied the role of endonuclease G (ENDOG) during autophagy. Using various assays they showed ENDOG promotes autophagy by partially suppressing mTORC1. This suppression is triggered by the interaction of ENDOG with 14-3-3g, which is promoted by GSK-3 β -dependent phosphorylation on ENDOG. Data in this manuscript are interesting and solid, and well support the authors' claims. Although the assays to probe the interaction between ENDOG and 14-3-3g were performed under massively over-expressing conditions and could be improved by using

endogenous antibody (if available), I think this is a beautiful piece of work and support publication on Nature Communications.

Some suggestions that the authors may want to consider after this manuscript is published:

1. Does manipulating the protein level of ENDOG alter the metabolite concentration in the cell or in the lysosome? The authors may use mass-spec with Lysosome-IP to probe this question.
2. Are TSC2 and VPS34 the only two proteins that are competed away by ENDOG? The authors may use quantitative proteomics to evaluate other proteins that bind to 14-3-3 and regulate mTORC1 activity.

Point-by-point response

We appreciate the Editor and Reviewers for considering the strengths of our work and for their valuable advice and suggestions for improving this manuscript. We have tried our best to address these points by conducting new experiments and revising the manuscript. Below are our point-by-point responses (*blue italic font*) to the reviewers' comments.

Reviewers' comments:

Reviewer #1 (Remarks to the Author):

General Comments:

This manuscript, entitled "Endonuclease G promotes autophagy by suppressing mTOR signaling and activating the DNA damage response", investigated the function and underlying mechanisms of ENDOG in regulating autophagy. Authors tried to conclude that ENDOG promotes autophagy during starvation by phosphorylation-mediated interaction with 14-3-3g, and its endonuclease activity-mediated DNA damage response. The major concern of this manuscript is lack of novelty. EndoG functions as a paternal mitochondria degradation factor through autophagy machinery, which has been reported in 2016. All the mTOR and autophagy pathways studied in this manuscript have been well established/predicted in the field already. Authors mainly validate the pathways in this manuscript. The rationale of this study is not strong. Moreover, the evidence for the key conclusion that endoG promotes autophagy is not convincing. This study also heavily relies on the artifact overexpression system. Many important conclusions were made based on the artifact overexpression system. Lots of other concerns are shown as following.

Major Comments:

1. Authors concluded that endoG promotes autophagic flux mainly based on the increase of LC3B-II in endoG over-expression cells. However, LC3B-II is not a specific marker for autophagy, as recent studies showed its role in phagocytosis (Cunha LD, et al., Cell 2018)

and endocytosis (Heckmann BL et al., Cell 2019). Additional specific autophagy markers should be tested in order to support authors' conclusion.

Answer: We agree with the reviewer that the function of LC3 is not only for autophagy, but also for phagocytosis and endocytosis. As suggested, we performed more experiments to support our conclusion that ENDOG promotes autophagy. We detected autophagic vesicles using electronic microscopy in the ENDOG knockout cells and ENDOG knockout mice. The EM images showed that loss of ENDOG reduced number of autophagic vesicles in L02 cell and in mice livers (Figures 1H-I and 2C-D). Moreover, loss of ENDOG repressed the protein expression of p-Becn1, p-ATG13, ATG14 and ATG12, which participate in the initiation or elongation of autophagy process (Figure S1E-F).

All these data consistently demonstrated that ENDOG indeed promoted autophagy.

Figure 1H-I

H-I. Representative electron microscopic images (H) and quantitative results (I) of autophagic vesicles in wild type or ENDOG knockout cells after treated with 100 nM BafA1 for 6 hours (red arrow: autophagic vesicle, AL: autolysosomes, AP: autophagosomes; n = 10 independent cells)

Figure 2C-D

C

D

C-D. Representative electron microscopic images and quantification of autophagic vesicles in *Endog*^{+/+} or *Endog*^{-/-} mouse livers after starvation for 24 hours (LD: lipid drop; N: nuclear; red arrow: autophagic vesicle; n = 10 independent fields, 2-3 cells per field)

Figure S1E-F

E-F. Representative western blots and quantitative results of autophagy related proteins in wild type or ENDOG knockout cells (n = 4 independent samples) (* p < 0.05; ** p < 0.01; *** p < 0.001)

2. Fig. 1A, why the expression of LC3B-I and LC3B-II was much higher in endoG-overexpressing cells than pk-Myc cells? Why there was no expected reduction on LC3B-I after it was converted into LC3B-II? Authors should exclude the possibility that was caused by the unequal loading.

Answer: We repeated the experiments by overexpressing ENDOG in different cell lines and found that ENDOG overexpression could elevate the expression of both LC3B-I and LC3B-II in L02, HepG2 and MHCC97-H cells (As showed in below Figure B-D). These results were highly reproducible in our hands.

We also checked the transcriptional level of LC3B, and found that ENDOG has no effect on the mRNA levels of LC3B (As showed in below Figure A). It has been reported that, the total amount of Atg8/LC3 homologue (Aut7p) was increased when autophagy was induced in yeast (Wei-Pang Huang, et, al. 2000 J Biol Chem), suggesting that LC3B-I and LC3B-II can be increased simultaneously during autophagy. In mammalian cells, the amount of LC3-II, the LC3-II/LC3-I ratio or LC3-II/ (LC3-I + LC3-II) ratio is used as autophagy indicators. However, due to the clear correlation of LC3B-II with autophagosome numbers, comparison of the amount of LC3B-II among samples is a more accurate indicator of autophagy (Noboru Mizushima, et, al. 2007 Autophagy). Thus, in this study we use the amount of LC3B-II as an autophagic indicator.

ENDO G promotes autophagy in several hepatocyte cell lines. A mRNA levels of *ENDO G* and *MAP1LC3B* in hepatocytes after overexpression of *ENDO G* for 48 hours. **B-D** Western blots of *ENDO G*, *LC3B* and *ACTB/GAPDH* in hepatocytes after overexpression of *ENDO G* for 48 hours. (***) $p < 0.001$)

3. Fig. 1B, it is not clear what and why is the strong green signal in the nucleus on the left panel (pk-Myc). There is no sufficient data explanation. The results of this part as well as many other parts were not clearly described.

Answer: Although LC3 is thought to function in the autophagosome formation, which may localize in the cytosol, the GFP-LC3 signaling was detected both in the nucleus and cytoplasm in many reports (Zhaoyang Li, et, al. 2019 Nature; Cefan Zhou, et, al. 2019 Autophagy). On principle, given the low molecular weight (about 18 KD) of LC3, GFP-LC3 could potentially enter the nucleus by passively diffusion even fused with GFP (about 27 KD) (Kimberly R. Drake, et, al. 2010 Plos One; Laura J. Terry, et, al. 2007 Science). The nuclear

enrichment of GFP-LC3 is independent of the autophagy induction (Kimberly R. Drake, et, al. 2010 Plos One).

4. Fig. 1E, why two repeats have so different results-SQSTM1?

Answer: SQSTM1 encodes the cargo adaptor protein, p62, which interacts with autophagic substrates and delivers them to autophagosomes for degradation. However, p62 itself is thought to be dispensable for canonical autophagy. In contrast to almost all of the core ATG proteins whose loss in mice results in embryonic or neonatal lethality, the $p62^{-/-}$ mice is mature-onset obesity (Akiko Kuma. et, al.2017 Autophagy). Moreover, recent publications have shown that p62 is a multifunctional scaffolding protein that interacts with a variety of proteins to regulate diverse processes including apoptosis, necroptosis, as well as redox state via regulation of the KEAP1-NRF2 pathway (Pablo Sánchez-Martín, et, al.2018, J Cell Sci). These diverse roles suggest that homeostatic regulation maybe affect p62 levels independent of autophagy. Thus, the expression of SQSTM1 may viable for some unknown reasons. We repeated this experiment carefully and revised the SQSTM1 results in the Figure 1J.

Figure1J

J

Western blots of LC3B, SQSTM1 and ENDOG in wild-type cells (WT), ENDOG knockout (KO) cells upon starvation treatments

5. Fig. 1F, authors compared the protein levels in CT and BafA1 groups on two separated blots. These two groups have to be run on the same blot in order to do such comparison.

Answer: Sorry for the confusion. In our original images, we did load these two groups in the same gel and performed western blot on the same blot, as shown in the following image.

To further clarify this issue, we reproduced these results, as showed in the revised Figure 1L-M

Figure 1L-M

L-M: Western blots and quantitative results of LC3B, SQSTM1 and ENDOG in wild-type cells (WT), ENDOG knockout (KO) cells upon and BafA1 (L-M) treatments (n = 4 biologically independent samples) (Scale bar = 10 μ m; * p < 0.05; ** p < 0.01; *** p < 0.001).

6. Fig. S1A, no reduction was observed with SQSTM1. Again, why overall expression of LC3B-I and II was increased? Why there was no expected reduction of LC3B-I, when it was converted into LC3B-II?

Answer: In our original figure, the reduction of SQSTM1 was mild but statistically significant. We repeated the experiment and found ENDOG expression indeed caused reduction of SQSTM1. To better demonstrate the reduction, we replaced the SQSTM1 band in Figure S1A.

The question about the expression of LC3B-II. Please refer to the answer for Question 2.

Figure S1A-B

A-B. *Representative western blots and quantitative results of autophagy related proteins (transfected with pK-Myc or ENDOG for 48 hours; n = 3 independent samples)*

7. EndoG knockout mice information was not provided in this manuscript. It was not clear how the knockout mice were made and how old the mice were used in the experiments.

Answer: The Endog^{+/-} and Endog^{-/-} mice used in this study were 3-month old male mice. We added this information in the "Methods and Materials-Autophagy analysis in mice, C. elegans and Drosophila" part. The information about how the Endog knockout mice were made was provided in the "Methods and Materials-Animals" part in the revised manuscript.

8. Line 111, authors claimed "showed decrease LC3B accumulation". However, no obvious difference was observed in Fig. 2A to support authors' conclusion.

Answer: In our original figure, the decrease of LC3B was statistically significant (Figure 2A), though the blot itself might not look so obvious.

A

To further address this concern, we repeated this experiment and got the same result that the level of LC3B-II in *Endog*^{-/-} mice livers were lower than that in the *Endog*^{+/+} livers (revised Figure 2A).

Figure 2A

A-B. Western blots and quantification of LC3B and SQSTM1 in *Endog*^{+/+} or *Endog*^{-/-} mouse livers after starvation for 24 hours ($n = 7$ biologically independent animals)

Furthermore, the electron microscopic images also showed fewer autophagic vesicles in the *Endog*^{-/-} mice livers under starvation treatment (revised Figure 2C-D).

These data suggested that loss of ENDOG repressed starvation-induced autophagy in mouse liver.

Figure 2C-D

C-D. Representative electron microscopic images and quantification of autophagic vesicles in *Endog*^{+/+} or *Endog*^{-/-} mouse livers after starvation for 24 hours (LD: lipid drop; N: nuclear; red arrow: autophagic vesicle; n = 10 independent field, 2-3 cells in per field)

9. Authors concluded that 14-3-3 γ interacts with endoG, which was not convincing. It was only observed in an artifact overexpression system. Authors also claimed that Starvation enhanced their interaction, which was also not true. No specific interaction signal was observed in Fig. 3J.

Answer: We performed new endogenous Co-IP experiment using better secondary antibody (Abcam 131366). The results showed that ENDOG binds with 14-3-3 γ even in the normal conditions, starvation enhanced the binding between ENDOG and 14-3-3 γ (revised Figure 3J). Meanwhile, the interaction between TSC2/Vps34 and 14-3-3 γ were weakened. These endogenous Co-IP data suggested that ENDOG and 14-3-3 γ have an interaction.

Figure 3J

J. Endogenous Co-IP experiments showed that starvation enhances the interaction between ENDOG and 14-3-3 γ but weaken the interaction of TSC2/Vps34 with 14-3-3 γ (in L02 WT cell; Star.: starvation for 12 hours; S: short time exposure; L: long time exposure).

10. The link between DNA damage response and endoG-mediated autophagy was rather weak. It was not clear what was the rationale to use etoposide to damage DNA and how it could be linked to starvation-induced autophagy. It was not a surprise to see the involvement of endoG in DNA damage response according to previous studies. However, this study did not show how endoG was involved in DNA damage response. Authors claimed that endoG mediated DNA damage response by its endonuclease activity. However, there is no any direct evidence to support this conclusion. It is mandatory to use its endonuclease deficient mutant to confirm their conclusion.

Answer: DNA damage has been reported as an early event during the starvation induced autophagy, the early DNA damage promotes autophagy through the PARP-1/AMPK or ATM/CHK2 pathway (José Manuel Rodríguez-Vargas, et, al. 2012 Cell Research). In the present study, we found ENDOG enhanced autophagy both under the mild DNA damage (starvation treatment) and strong DNA damage (etoposide treatment). ENDOG-mediated DNA damage repressed the mTOR pathway and promoted autophagy initiation, which finally promoted autophagy flux.

*We treated the wild-type and ENDOG knockout cells for HBSS for a shorter time (1 or 3 hours, shorter than 6 or 12 hours in the previous manuscript). The comet assay showed that starvation caused DNA damage in the early time, and the loss of ENDOG repressed the starvation induced DNA damage (revised Figures 5A-C). The p-H2A.X foci staining results demonstrated that DNA damage was accumulated over time in the wild-type cells, while the accumulation was slower in the ENDOG knockout cells (Figures 5D-E). **These data suggested that DNA damage occurred during the starvation-induced autophagy, and ENDOG participated in the starvation-induced DNA damage at an early point.***

Figure 5:

Figure 5. Loss of ENDOG repressed the starvation induced DNA damage and autophagy. A-C. Representative images of comet assay in wild-type or ENDOG knockout L02 cells (A) and the quantification of tail DNA (B) and tail moment (C) (starvation treated for 1 or 3 hours; $n = 75-100$ independent cells). D-E. Representative images of p-H2A.X foci (D) and quantitative results (E) in wild-type or ENDOG knockout (KO) L02 cells at the indicated time point after the starvation treatment (Scale bar = 10 μm , $n = 50$ independent cells). F-I. Western blots and quantitative results of the indicated proteins in wild type or ENDOG knockout L02 cells following starvation treatment ($n = 4$ independent samples) (* $p < 0.05$; ** $p < 0.01$; *** $p < 0.001$; **** $p < 0.0001$).

Starvation promoted the activation of PARP1-AMPK axis, which repressed the

mTOR activity and finally induced autophagy (José Manuel Rodríguez-Vargas, et, al. 2012 *Cell Research*). Here, we found that loss of *ENDOG* repressed both expression and activation of *PARP1*, as well as activation of *AMPK* both under normal and starvation conditions (Figures 5F-G). Consistently, we found that loss of *ENDOG* promoted *mTOR* signaling and repressed autophagy (Figures 5F-G). Another report demonstrated starvation activated *CHK2* to phosphorylate *Becn1* that promoted autophagy induction (Qi-Qiang Guo, et, al.2020 *The EMBO Journal*). In the present study, we found loss of *ENDOG* significantly repressed the starvation induced activation of both *CHK1/2* and autophagy (Figures 5H-I). All these data suggested that under starvation, *ENDOG* induced DNA damage promoted autophagy.

ENDOG is an endonuclease which cleaves DNA during apoptosis, we hypothesized that *ENDOG* may cause DNA damage. We indeed found that *ENDOG* overexpression caused DNA damage (Figure 6). The comet assay revealed that *ENDOG* could cause DNA fragmentation under normal condition and excessive DNA fragmentation when treated with etoposide (Figures 6A-C). These data suggested that overexpression of *ENDOG* may activate the DNA damage by causing the DNA fragmentation.

Figure 6A-C

A-C. Representative images of comet assay in wild type or *ENDOG* overexpressed L02 cells (A) and the quantification of tail DNA (B) and tail moment (C) (WT: wild type L02; OE: *ENDOG* overexpressed L02; Eto.: 50 μM etoposide for 1 hour; n = 75-150 independent cells).

Previous studies have reported that overexpression of *ENDOG* promoted DNA fragmentation and cell death, while the inactivated form of *ENDOG* (mutation of the catalytic activity) could not (Schafer, P., et al. 2004 *J Mol Biol*). In our study, we mutated the H-N-N (141H/163N/173N) motif of *ENDOG* to make an endonuclease activity deficient form of *ENDOG* (EM-*ENDOG*). We found that the wild-type *ENDOG* could induce DNA fragmentation, while EM-*ENDOG* could not (FiguresS13A-C).

Figure S13A-C:

A-C. Representative images of comet assay (A) and the quantification of tail DNA (B) and tail moment (C) (ENDO G knockout cells were transfected with pK-Myc, wild-type and EM-ENDOG for 48 hours; $n = 75-150$ independent cells).

The p-H2A.X foci assay also demonstrated that compared to the wild type ENDOG, the EM-ENDOG has a lower DNA damage and weaker DNA damage response (Figures S12D-F). Moreover, we treated wild-type cells with ENDOG specific inhibitor PNR-3-80 to find that ENDOG inhibition repressed the etoposide induced DNA damage and autophagy (Figure S14).

These data suggested that the endonuclease activity of ENDOG was necessary for ENDOG-induced DNA damage and autophagy.

Figure S14

Figure S14. Inhibition of ENDOG activity represses DNA damage-induced autophagy. A-B. Representative images (A) and respective quantitative results (B) of p-H2A.X foci in L02 cells ($n = 15-20$ independent fields; 5-8 cells per field). C-D. Representative images (C) and respective quantitative results (D) of GFP-LC3 puncta in L02 cells ($n = 10$ independent fields; 5-8 cells per field). E-F. Western blots (E) and quantification (F) of the indicated proteins ($n = 4$ independent samples). (PNR-3-80: ENDOG inhibitor, $50 \mu\text{M}$ for 24 hours; Eto.: etoposide, $50 \mu\text{M}$ for 1 hour; Scale bar = $10 \mu\text{m}$; * $p < 0.05$; ** $p < 0.01$; *** $p < 0.001$).

11. Fig. S9, p-H2Ax should be checked at multiple time points. 1 h after etoposide is a little bit to short usually.

Answer: As recommended, we treated different cell groups (WT, ENDOG KO, ENDOG KO + ENDOG, ENDOG KO + Δ 1-48, ENDOG KO + EM, ENDOG KO + ENDOG-NLS) with $50 \mu\text{M}$ etoposide for 1 hour with recovery for 1, 3, 6 and 12 hours. The data again showed EM-ENDOG has little activity in DNA damage. While, Δ 1-48 and ENDOG-NLS forms could induce DNA damage in ENDOG knockout cells (revised Figures S12D-E). Furthermore, the western blots also demonstrated the EM form of ENDOG has a weaker DNA damage response (revised Figure S12F).

Figure S12D-F

D-E. Representative images of p-H2A.X foci (E) and quantitative results (F) in the indicated cell groups at different time points after the etoposide treatment ($n = 50$ independent cells). **F.** Western blots of the indicated proteins in wild type, ENDOG KO and ENDOG KO cells transfected with the ENDOG mutants. (Scale bar = $10\mu\text{m}$; *** $p < 0.001$).

Minor Comments:

1. Grammar errors. For example, "A previous results..." on page 4, line 63.

Answer: Thanks for pointing out this typo. We have corrected it in the revised manuscript.

2. Fig. 1, it is better to label each panel, including quantification panels.

Answer: We have labeled each panel as suggested.

3. Fig. S2A, what is the difference between the left and right panels? The label is missing.

Answer: The left panels in Fig. S2A were experiments performed in the WT L02 cells, and the right panel were performed in the ENDOG overexpression cells. We revised the label accordingly.

Reviewer #2 (Remarks to the Author):

This manuscript reported that the Endonuclease G (ENDOG) promotes autophagy through inhibiting mTOR and activating DDR. Mechanistically, ENDOG is phosphorylated by GSK-3b, which enhance its interaction with 14-3-3, leads to the release of TSC2 and Vps34, and inhibit mTOR. The author also showed that ENDOG promotes DDR initiation. However, the molecular mechanism is still not clear. For example, how ENDOG activate DDR? How EM mutant affect mTOR signaling? And some data is not in high quality. There are currently a number of shortcomings with the experimentation I listed below that should be addressed in order to strengthen the findings.

1. In Fig 3I,J, for the Co-IP data, the blot for TSC2(I) and ENDOG(J) is in low quality. Please repeat this experiment and show better blot data.

Answer: As suggested, we have repeated this experiment and replaced better blots in Figure 3I and 3J.

Figure 3I-J

I. Co-IP experiments showed that ENDOG overexpression decreases interactions between 14-3-3 γ with TSC2/Vps34 (L02 ENDOG-KO cell co-overexpressed Myc-14-3-3 γ with Flag-ENDOG or empty vector for 48 hours). J. Endogenous Co-IP experiments showed that starvation enhances the interaction between ENDOG and 14-3-3 γ but weaken the interaction of TSC2/Vps34 with 14-3-3 γ (in L02 WT cell; Star.: starvation for 12 hours; S: short time exposure; L: long time exposure).

2. In Fig 3K, The LC3B level of ENDOG+14-3-3group is higher than 14-3-3group. But in Fig 3M, they are similar. The author should explain the inconsistent results.

Answer: The blot shown in the original submitted figure was in low quality. We repeated this experiment and confirmed that the LC3B levels of ENDOG+14-3-3 group and 14-3-3 group are similar. We replaced the figure with a higher quality one. Additionally, we overexpressed ENDOG and 14-3-3 γ

together or respectively, and then treated cells with starvation. Under starvation, 14-3-3 γ weakened ENDOG induced mTOR repression (Figure S6). All these data suggested 14-3-3 γ overexpression could partially block ENDOG induced autophagy.

Figure S6

Figure S6. 14-3-3 γ repressed the ENDOG induced mTOR repression and autophagy. Western blots of the indicated proteins (ENDOG or 14-3-3 γ were transfected for 48 hours; HBSS treated for 6 hours; S: short exposure; L: long exposure)

3. In Fig 4, due to mTOR activation, The LC3B level of ENDOG-AA should be decreased compared to ENDOG-WT and DD groups. But the blot result is opposite and it is not match with the quantification data. The author should explain that and check the data more carefully.

Answer: We apologize for the mistake that we showed a wrong band for LC3B., and now replace it with the right one. Meanwhile, we overexpressed wild-type ENDOG, ENDOG-AA and ENDOG-DD in ENDOG knockout cells and treatment with BafA1, and found that only overexpression of wild-type ENDOG and ENDOG-DD, but not the AA-form could enhanced LC3B-II accumulation (Figure S8), suggesting the phosphorylation of Thr 128 and Ser 288 are necessary for ENDOG-induced autophagy.

FigureS8

Figure S8. Phosphorylation of T128 and S288 is necessary for ENDOG mediated autophagy. A-B. Western blots of the indicated proteins (A) and the quantification of LCEB-II (B) (ENDOG KO cells transfected with pk-Myc, wild type ENDOG, ENDOG-DD and ENDOG-AA for 48 hours; BafA1: 100 nM, for 6 hours; S: short exposure; L: long exposure; n = 4 independent samples; * p < 0.05; ** p < 0.01)

4. In Fig 4J, the label is wrong. That should be ENDOG-AA but not DD.

Answer: Thanks for pointing out this typo, we have revised it.

5. In Fig 5G-H, with ATM inhibition, overexpression of ENDOG cannot increase LC3 punta. However, overexpression of ENDOG should still activate autophagy through competing 14-3-3 binding. Does that mean ATM also affect GSK-3 β signaling? The author should perform further experiment to confirm the result.

Answer: Firstly, we investigated whether ATM affected the GSK-3 β pathway. Here we used the etoposide to activate ATM and KU6009 to inhibit ATM in L02 cells. The results showed that under the etoposide treatment, ATM was activated and the phosphorylation of GSK-3 β Ser 9 was increased. When treated with the ATM specific inhibitor KU60019, phosphorylation of GSK-3 β ser 9 was repressed both in presence or absence of the etoposide (As in the figure below A-B). These data suggest that, in our model, KU60019 may repress phosphorylation of GSK-3 β ser 9, which was consistent with the previous work (Yuka Hotokezaka, et al. 2020 Commun Biol). As phosphorylation of GSK-3 β ser 9 is the inactivated form GSK-3 β , KU60019 may activate the GSK-3 β signaling by repressing the phosphorylation at Ser 9.

KU60019 treatment affects GSK-3 β pathway and the binding between ENDOG and 14-3-3 γ . A-B. Western blots (A) and the quantification results (B) of the indicated proteins (L2 cells were treated with the 10 μ M KU60019 or 50 μ M etoposide for 1 hour). C. Co-IP experiment (Flag-ENDOG stably overexpressing L2 cells were treated with 10 μ M KU60019 for 1 hour).

Moreover, KU60019 treatment decreased the phosphorylation of ENDOG and the interaction between ENDOG and 14-3-3 γ (as showed in figure C). These data suggested that KU60019 might repress the activity of other unidentified kinases that could phosphorylate ENDOG. This needs to be further studied in the future.

Taken together, on one hand Ku60019 treatment blocked ENDOG induced DNA damage, on the other hand it blocked the interaction of ENDOG with 14-3-3 γ and eventually suppressed autophagy. That may explain why KU60019 represses autophagy even in the ENDOG overexpressed cells.

On the other hand, overexpression of ENDOG increased γ -H2AX foci with ATM inhibition. What is the mechanism?

Answer: Our results demonstrated that ENDOG induces DNA damage. After DNA strand breaks, ATM, ATR and DNA-PK are activated. We found that KU60019, ATM specific inhibitor, only repressed the phosphorylation of ATM, but has no effect on the phosphorylation of ATR (Figure S11), suggesting that ENDOG-mediated DNA damage may increase p-H2A.X foci through the ATR in the presence of KU60019.

Figure S11

Figure S11. KU60019 treatment partially repressed the ENDOG induced DNA damage. A-B. Western blots (A) and quantification (B) of the indicated proteins in wild type and ENDOG overexpressed cells following the KU60019 treatment. (KU60019: 10 μ M for 1 hour; n = 4 independent samples; * p < 0.05; ** p < 0.01; *** p < 0.001)

6. From Fig 5, it looks like that ENDOG promotes DDR initiation, such as p-ATM and γ -H2AX at early time point. What is the mechanism? At least, the author should test the different ENDOG mutants (listed in Fig 6A) function in DDR activation.

Answer: As an endonuclease, ENDOG causes DNA double- or single-stranded breaks (A Ruiz-Carrillo, et, al. 1987 EMBO J). The comet assay result showed that ENDOG promotes DNA fragmentation (Figures 6A-C), which could activate the downstream DNA damage response (DDR).

Figure 6A-C

A-C. Representative images of comet assay in wild type or ENDOG overexpressed L02 cells (A) and the quantification of tail DNA (B) and tail moment (C) (WT: wild type L02; OE: ENDOG overexpressed L02; Eto.: 50 μ M etoposide for 1 hour; n = 75-150 independent cells).

Besides, we found that the expression and cleavage of PARP1 (a classical DNA damage sensor) was significantly repressed in ENDOG knockout cells

(Figures 5F-G). These data suggested that ENDOG promoted PARP-1 expression and activation in response to DNA damage.

Figure 5F-G

F-G. Western blots and quantitative results of the indicated proteins in wild type or ENDOG knockout L02 cells following starvation treatment ($n = 4$ independent samples) (* $p < 0.05$; ** $p < 0.01$; *** $p < 0.001$; **** $p < 0.0001$).

As suggested, we overexpressed these ENDOG mutants into the ENDOG knockout cells and treated them with etoposide. We found that the wild-type and NLS-ENDOG, but not the Del 1-48 and EM forms of ENDOG, had increased DNA damage in the ENDOG knockout cells (Figures S12A-C). The p-H2A.X foci staining also showed that the Del 1-48 and EM-ENDOG had weaker DNA damage response after etoposide treatment (Figures S12D-E). Moreover, the expression of DNA damage sensor (PARP-1) and other DNA damage response proteins (p-ATM, p-ATR, p-CHK1, p-CHK2 and p-H2A.X) in Del 1-48 and EM-ENDOG groups were less than that in the wild-type and ENDOG-NLS groups (Figure S12F).

Figure S12. Endonuclease activity of ENDOG is essential for the DNA damage response and autophagy induction. A-C. Representative images (A) and respective quantitative results (B) of GFP-LC3 puncta and p-H2A.X foci under etoposide treatment (C) (WT: wild-type; KO: ENDOG knockout; plasmids transiently transfected for 48 hours; etoposide: 50 μ M for 1 hour; n = 10-15 independent fields; 5-8 cells per field). D-E. Representative images of p-H2A.X foci (E) and quantitative results (F) in the indicated cell groups at

different time points after etoposide treatment ($n = 50$ independent cells). *F.* Western blots of the indicated proteins in wild-type, ENDOG KO cells, and ENDOG KO cells transfected with the ENDOG mutants (Scale bar = 10 μm ; *** $p < 0.001$).

To further confirm that the DNA endonuclease activity is necessary for ENDOG-induced DNA damage and autophagy, we overexpressed the wild-type and EM-ENDOG in the ENDOG knockout cells. The comet assay results showed that EM-ENDOG could not induce DNA damage in the ENDOG knockout cell (Figures S13A-C). Furthermore, we found that compared to the wild-type ENDOG, the EM-ENDOG has less expression and activation of PARP-1, as well as less activation of AMPK and TSC2 both in the normal and starvation conditions (Figures S13D-E). Consistently, the mTOR activity in EM-ENDOG group is higher than that of the wild-type ENDOG group (Figures S13D-E), suggesting that EM-ENDOG lost the ability to repress the mTOR. Furthermore, PNR-3-80, a specific ENDOG inhibitor, could repress the etoposide-induced DNA damage and autophagy (Figure S14).

Taken together, these data suggested that ENDOG mediated DNA damage through its endonuclease activity.

Figures S13

Figure S13. Endonuclease activity of ENDOG is essential for ENDOG mediated DNA damage and mTOR repression under the starvation. A-C. Representative images of comet assay (A) and the quantification of tail DNA (B) and tail moment (C) (ENDOG

knockout cells were transfected with pK-Myc, wild-type and EM-ENDOG for 48 hours; $n = 75-150$ independent cells). D-E. Western blots (D) and quantification (E) of the indicated proteins (ENDOG knockout cells were transfected with pK-Myc, wild-type and EM-ENDOG for 48 hours and treated with HBSS for 6 hours; $n = 4$ independent samples; S: short exposure; L: long exposure; * $p < 0.05$; ** $p < 0.01$; *** $p < 0.001$)

Figure S14

Figure S14. Inhibition of ENDOG activity represses DNA damage-induced autophagy. A-B. Representative images (A) and respective quantitative results (B) of p-H2A.X foci in L02 cells ($n = 15-20$ independent fields; 5-8 cells per field). C-D. Representative images (C) and respective quantitative results (D) of GFP-LC3 puncta in L02 cells ($n = 10$ independent fields; 5-8 cells per field). E-F. Western blots (E) and quantification (F) of the indicated proteins ($n = 4$ independent samples). (PNR-3-80: ENDOG inhibitor, 50 μM for 24 hours; Eto.: etoposide, 50 μM for 1 hour; Scale bar = 10 μm ; * $p < 0.05$; ** $p < 0.01$; *** $p < 0.001$).

7. In Fig 6, the author suggested that EM mutant cannot rescue LC3B punta because of its activation of mTOR. My question is how EM mutant (endonuclease activity deficient) affect its binding with 14-3-3 and activate mTOR? Does it affect GSK3-beta phosphorylation on ENDOG? Also, does EM mutant affect DDR? All these are important question to clarify the mechanism that how ENDOG activate autophagy.

Answer: EM-ENDOG lost the DNase function, it could not initiate DDR, and thus EM ruined ENDOG's autophagy promotion ability.

In ENDOG knockout cells, overexpression of wild-type ENDOG induced DNA fragmentation, while the EM-ENDOG failed to (Figures S13A-C). Compared to the wild-type, EM-ENDOG induced a weaker expression and activation of PARP1 (Figures S13D-E), accompanied with lower expressions of p-AMPK and p-TSC2, but higher expression of p-mTOR, p-ULK1, p-p70S6K and p-4EBP1 (Figures S13D-E). These data together indicate that EM-ENDOG activated the mTOR pathway by repressing the PARP1-AMPK axis.

Figures S13

Figure S13. Endonuclease activity of ENDOG is essential for ENDOG mediated DNA damage and mTOR repression under the starvation. A-C. Representative images of comet assay (A) and the quantification of tail DNA (B) and tail moment (C) (ENDOG knockout cells were transfected with pK-Myc, wild-type and EM-ENDOG for 48 hours; $n = 75-150$ independent cells). D-E. Western blots (D) and quantification (E) of the indicated proteins (ENDOG knockout cells were transfected with pK-Myc, wild-type and EM-ENDOG for 48 hours and treated with HBSS for 6 hours; $n = 4$ independent samples; S: short exposure; L: long exposure; * $p < 0.05$; ** $p < 0.01$; *** $p < 0.001$)

Besides, we overexpressed the wild-type and EM-ENDOG in the ENDOG knockout cells, and found that wild-type ENDOG enhanced the DNA damage response, while the expression level of p-ATM, p-ATR, p-CHK1, p-CHK2 and

p-H2A.X had little change in the EM-ENDOG overexpression group (Figure S12F), suggesting that EM-ENDOG could not activate DDR in the ENDOG knockout cells.

Figure S12F

F. Western blots of the indicated proteins in wild type, ENDOG KO and ENDOG KO cells transfected with the ENDOG mutants.

The Co-IP experiment results showed that the phosphorylated EM-ENDOG represented a much smaller portion in the total EM-ENDOG, which might compromise the binding with 14-3-3 γ (as showed in below figure A) and thus decreased its inhibition to mTOR, which also ruined ENDOG's autophagy promotion ability.

Combined these two effects together, EM-ENDOG could not rescue LC3B puncta in ENDOG knockout cells.

A. Co-IP results showed that endonuclease activity mutant did not affect the phosphorylation of ENDOG and the binding between ENDOG and 14-3-3 γ (ENDOG knockout cells were transfected with ENDOG-Myc / EM-ENDOG-Myc and GSK-3 β for 48 hours).

To further examine the ENDOG's function to induce DDR to promote autophagy, we treated cells with ENDOG specific inhibitor, PNR-3-80 which could inhibit ENDOG's nuclease activity (Jang, et, al. 2015 DNA Cell Biol) but did not affect ENDOG's binding with 14-3-3 γ (as showed in below figure B).

B. Chemical inhibition of ENDOG activity did not affect the phosphorylation of ENDOG and the binding between ENDOG and 14-3-3 γ (Flag-ENDOG stably overexpressing L02 cells were treated with PNR-3-80 for 24 hours)

Moreover, the results showed that PNR-3-80 could repress DNA damage and autophagy (Figure S14), suggestion the role ENDOG played in autophagy through the DDR pathway.

Figure S14

Figure S14. Inhibition of ENDOG activity represses DNA damage-induced autophagy. A-B. Representative images (A) and respective quantitative results (B) of p-H2A.X foci in L02 cells (n= 15-20 independent fields; 5-8 cells per field). C-D. Representative images (C) and respective quantitative results (D) of GFP-LC3 puncta in L02 cells (n= 10 independent fields; 5-8 cells per field). E-F. Western blots (E) and quantification (F) of the indicated proteins (n = 4 independent samples). (PNR-3-80: ENDOG inhibitor, 50 μ M for 24 hours; Eto.: etoposide, 50 μ M for 1 hour; Scale bar = 10 μ m; * p < 0.05; ** p < 0.01; *** p < 0.001).

Taken together, ENDOG promotes autophagy through the suppression of mTOR by its phosphorylation-mediated interaction with 14-3-3 γ and its endonuclease activity-mediated DNA damage response.

Reviewer #3 (Remarks to the Author):

In the manuscript, the authors studied the role of endonuclease G (ENDOG) during autophagy. Using various assays they showed ENDOG promotes autophagy by partially suppressing mTORC1. This suppression is triggered by the interaction of ENDOG with 14-3-3g, which is promoted by GSK-3b-dependent phosphorylation on ENDOG. Data in this manuscript are interesting and solid, and well support the authors' claims. Although the assays to probe the interaction between ENDOG and 14-3-3g were performed under massively over-expressing conditions and could be improved by using endogenous antibody (if available),

As suggested, we repeated the endogenous Co-IP experiment using better antibody and confirmed the interaction of ENDOG and 14-3-3 γ .

J. Endogenous Co-IP experiments showed that starvation enhances the interaction between ENDOG and 14-3-3 γ but weakens the interaction of TSC2/Vps34 with 14-3-3 γ (in L02 WT cell; Star. : starvation for 12 hours; S: short time exposure; L: long time exposure).

I think this is a beautiful piece of work and support publication on Nature Communications.

Some suggestions that the authors may want to consider after this manuscript is published:

1. Does manipulating the protein level of ENDOG alter the metabolite concentration in the cell or in the lysosome? The authors may use mass-spec with Lysosome-IP to probe this question.

Answer: Thanks for your kind suggestions. Due to the scope of our current work, we will investigate the metabolite and lysosome proteins in ENDOG knockout cells or mice by the mass-spec and metabolomics in the future.

2. Are TSC2 and VPS34 the only two proteins that are competed away by ENDOG? The authors may use quantitative proteomics to evaluate other proteins that bind to 14-3-3 and regulate mTORC1 activity.

Answer: As suggested, we performed new experiments to check whether there are other protein involved .We firstly investigated PRAS40, another mTOR suppressor, which has been shown to bind with 14-3-3 (Lifu Wang, et, al. 2008J Biol Chem).No interactionbetween14-3-3 γ and PRAS40 were detected .Additionally, we found ENDOG has little effect on the binding between 14-3-3 γ and Becn1. Furthermore, we also didn't detect the binding between 14-3-3 γ andPI3K, a kinase that activates the mTOR pathway through AKT. Next, to more accurately evaluate other proteins competing with 14-3-3 γ and regulating mTORC1 activity, we will use the quantitative proteomics in the future.

ENDOG knockout L02 cells were transfected with Flag-ENDOG and 14-3-3 γ -Myc for 48 hours.

REVIEWER COMMENTS

Reviewer #1 (Remarks to the Author):

Comments:

In this revision, authors did substantial new experiments and addressed some concerns. However, the major concerns were not addressed or completely ignored in this revision.

Here are the major concerns, which have been completely ignored: "The major concern of this manuscript is lack of novelty. EndoG functions as a paternal mitochondria degradation factor through autophagy machinery, which has been reported in 2016. All the mTOR and autophagy pathways studied in this manuscript have been well established/predicted in the field already. Authors mainly validate the pathways in this manuscript. The rationale of this study is not strong." Authors did not provide any new evidence to explain or support the novelty and rationale of this study.

In addition, the conclusion summarized in Abstract were not well supported by their data. For example, no convince data were provided to support "we report that ENDOG released from mitochondria promotes autophagy during starvation" or "GSK-3 β -mediated phosphorylation of ENDOG at Thr-128 and Ser-288 enhances its interaction with 14-3-3 γ , which leads to the release of TSC2 and Vps34 from 14-3-3 γ ". Does blocking endoG released from mitochondria or mutation of Thr-128 or Ser-288 reverse endoG effects?

Fig. 1 h figures are too small to show the typical autophagy morphology.

Fig. 2C-D. Is there special reason to compare EndoG $^{+/-}$ vs $^{-/-}$? In cells, authors compared WT vs KO. Why not mouse?

Fig. S1E-F.: Why Becn1, ATG13, 14, 5, 7 blots show no difference between WT and KO, but quantification shows significant difference? These data are not consistent.

In response to the previous comments "Fig. 1A, why the expression of LC3B-I and LC3B-II was much higher in endoG-overexpressing cells than pk-Myc cells? Why there was no expected reduction on LC3B-I after it was converted into LC3B-II? Authors should exclude the possibility that was caused by the unequal loading.", in this revision authors overexpressed endoG in different cell lines and found LC3B level was increased, which only partially answered the previous comment. It is known that LC3B levels are very sensitive to the culture conditions and stimulation, which can be induced by transfection. Here it lacks important controls to see if transfection itself increased LC-II level. Will overexpression of nuclease-inactive endoG increase LC3B levels? Rescue experiments are also mandatory to help confirm it is the endoG's specific effects.

In response to the previous comments "Fig. 1B, it is not clear what and why is the strong green signal in the nucleus on the left panel (pk-Myc). There is no sufficient data explanation. The results of this part as well as many other parts were not clearly described.", authors explained what and why is the strong green signal in the nucleus on the left panel. Meanwhile they provided the strong evidence of the disadvantage of the artifact over-expression system. Conclusions drew from this artifact system is a concern.

Fig. 2A: authors claimed "showed decrease LC3B accumulation". Again images show no difference, but statistically analysis with significance, which is strange and brought concerns. The conclusion is not convincing.

Fig. 3J: IgG controls are missing.

For the previous comment "The link between DNA damage response and endoG-mediated autophagy was rather weak. It was not clear what was the rationale to use etoposide to damage DNA and how it

could be linked to starvation-induced autophagy. It was not a surprise to see the involvement of endoG in DNA damage response according to previous studies. However, this study did not show how endoG was involved in DNA damage response. Authors claimed that endoG mediated DNA damage response by its endonuclease activity. However, there is no any direct evidence to support this conclusion. It is mandatory to use its endonuclease deficient mutant to confirm their conclusion.", authors indeed provided substantial data to confirm the effects of endoG on DNA damage response. However, no direct convincing data were provided in the revision to link between endoG-mediated DNA damage response with endoG-mediated autophagy.

Figure S14 E should be presented on the same blot.

Reviewer #2 (Remarks to the Author):

The author answered all my questions. I suggest to accept this manuscript for publication.

Point-by-point response

We appreciate Reviewer #1 for his valuable advice and suggestions for improving this manuscript. We have tried our best to address the remaining concerns by conducting new experiments and revising the manuscript. Below are our point-by-point responses (*blue italic font*) to the reviewers' comments.

Reviewer #1 (Remarks to the Author):

Comments:

In this revision, authors did substantial new experiments and addressed some concerns. However, the major concerns were not addressed or completely ignored in this revision. Here are the major concerns, which have been completely ignored: "The major concern of this manuscript is lack of novelty. EndoG functions as a paternal mitochondria degradation factor through autophagy machinery, which has been reported in 2016. All the mTOR and autophagy pathways studied in this manuscript have been well established/predicted in the field already. Authors mainly validate the pathways in this manuscript. The rationale of this study is not strong." Authors did not provide any new evidence to explain or support the novelty and rationale of this study.

Answer: We are sorry that we did not directly discuss the novelty issue in previous submission.

Actually, in 2016 Science paper, the authors showed that autophagy pathway and cps-6/EndoG seemed to act in parallel to mediate the paternal mitochondria elimination. However, they didn't show any data that cps-6/EndoG have function in autophagy, even not mentioned about it. This is why we continued study the role and underlying mechanism of ENDOG in autophagy in this study.

I am putting their original figure about the parallel function of lgg-1 and cps-6 in the following. Moreover, I am pasting the original description of the Science paper on the function of cps-6/ENDOG.

"Analyses of the double and the triple mutants among cps-6, lgg-1, and rad-23 indicate that cps-6, lgg-1, and rad-23 use distinct mechanisms (mitochondrial self-destruction, autophagy, and proteasomes, respectively) to coordinate swift and efficient PME." (Excerpted from Science, 2016, vol 353, Issue 6297, page 397, first paragraph of the middle column)

This figure is Fig.S6C of the Science paper, and it was clear that cps-6/ENDOG and lgg-1 (autophagy) likely acted in parallel.

Also, though mTOR signaling is a well characterized pathway in autophagy, we firstly reported ENDOG to be involved in mTOR signaling pathway to mediate autophagy. Its phosphorylation by GSK3 β and interaction with 14-3-3 γ were not reported yet before our findings.

In addition, the conclusion summarized in Abstract were not well supported by their data. For example, no convince data were provided to support "we report that ENDOG released from mitochondria promotes autophagy during starvation" or "GSK-3 β -mediated phosphorylation of ENDOG at Thr-128 and Ser-288 enhances its interaction with 14-3-3 γ , which leads to the release of TSC2 and Vps34 from 14-3-3 γ ". Does blocking endoG released from mitochondria or mutation of Thr-128 or Ser-288 reverse endoG effects?

Answer: These experiments are indeed very important for our paper, and we thank the reviewer for mentioning them. Actually, we did provide these data in our original and

revised submission. We showed that blocking ENDOG from mitochondria by Z-IETD-FMK could repress autophagy (Fig S15).

Fig S15:

Nonetheless, we performed more experiments to show that Z-IETD-FMK treatment could also reduce the endogenous LC3B puncta (revised Fig S15F).

For the "mutation of Thr-128 or Ser-288", we have also showed the results in our previous submission (Fig 4C-F and Fig S8). The results showed that, compared to the wild-type ENDOG and ENDOG-DD, overexpression of ENDOG-AA has less autophagosome numbers and LC3B-II accumulation under both the control and BafA1 treatment (Figures 4C-D and S8). Moreover, ENDOG-AA loses the ability to suppress mTOR pathway to promote LC3B-II accumulation and SQSTM1 degradation (Figures 4E-F).

Fig 4A-F:

Fig S8:

Fig. 1 h figures are too small to show the typical autophagy morphology.

Answer: As requested, we put a bigger picture in the revised manuscript, which also present as below.

Fig. 2C-D. Is there special reason to compare EndoG^{+/-} vs -/-? In cells, authors compared WT vs KO. Why not mouse?

Answer: For the mice, we use the littermates as control, since there was no obvious difference between the EndoG^{+/+} and EndoG^{+/-} mice. To gain enough EndoG^{-/-} mice, we crossed the heterozygote (EndoG^{+/-}) with homozygote (EndoG^{-/-}) mice, whose offspring were heterozygote (EndoG^{+/-}) or homozygote (EndoG^{-/-}) mice. So we used the heterozygous littermates as the controls, which also suggested by the Jaxson laboratory.

(<https://www.jax.org/jax-mice-and-services/customer-support/technical-support/breeding-and-husbandry-support/considerations-for-choosing-controls>).

Fig. S1E-F.: Why Becn1, ATG13, 14, 5, 7 blots show no difference between WT and KO, but quantification shows significant difference? These data are not consistent.

Answer: The expression levels of Becn1, ATG5 and ATG7 indeed had no differences between WT and the KO group in the blot, but our quantification results also showed no difference in the previous submission. In addition, the expression levels of ATG13 and ATG14 were reduced in the KO group.

The dot plots in the original bar chart were indeed confusing, so we redid the bar charts to label the p value level with "" or "n.s", as showed below.*

Fig. S1E-F:

In response to the previous comments "Fig. 1A, why the expression of LC3B-I and LC3B-II was much higher in endoG-overexpressing cells than pk-Myc cells? Why there was no expected reduction on LC3B-I after it was converted into LC3B-II? Authors should exclude the possibility that was caused by the unequal loading.", in this revision authors overexpressed endoG in different cell lines and found LC3B level was increased, which only partially answered the previous comment. It is known that LC3B levels are very sensitive to the culture conditions and stimulation, which can be induced by transfection. Here it lacks important controls to see if transfection itself increased LC-II level. Will overexpression of nuclease-inactive endoG increase LC3B levels? Rescue experiments are also mandatory to help confirm it is the endoG's specific effects.

Answer: In all the overexpression studies, we transfected the empty vector pK-Myc as the control. The transfection conditions were the same between pK-Myc and ENDOG groups. So it should not affect our conclusions.

As requested, we performed more experiments to include a new negative control of L02 cells without transfection. The western blots and endogenous LC3B immunofluorescent staining results showed that there was no difference between non-transfection group and pK-Myc group, while overexpression of ENDOG increased both LC3B-II accumulation and the number of LC3B dots per cell.

(A) Western blots of ENDOG, LC3B and GAPDH in L02 cells. (B) Immunofluorescent staining of endogenous LC3B in L02 cells. (L02 with no transfection or transfected with pK-Myc or ENDOG for 48 hours; Scale bar = 10 μ m).

For the rescue experiments, we had a few results in our revised manuscript. We did not put the image of western blots that evaluated the protein level of LC3B-II, as we set out to address how enzymatic mutations affected mTOR signaling. We included the level of LC3B-II here.

We evaluated the protein level of LC3B-II in the ENDOG-KO cells transfected with the wild-type ENDOG, nuclease-inactive ENDOG (EM-ENDOG), or empty vector plasmids. The results (Figure S13D-E) showed that EM-ENDOG could not induce the accumulation of LC3B-II, which consistently indicated that endonuclease activity of ENDOG is essential for autophagy induction in our revised manuscript.

Figure S13

In response to the previous comments "Fig. 1B, it is not clear what and why is the strong green signal in the nucleus on the left panel (pk-Myc). There is no sufficient data explanation. The results of this part as well as many other parts were not clearly described.", authors explained what and why is the strong green signal in the nucleus on the left panel. Meanwhile they provided the strong evidence of the disadvantage of the artifact over-expression system. Conclusions drew from this artifact system is a concern.

Answer: In these experiments, we co-transfected the GFP-LC3 plasmid with the pk-Myc or ENDOG plasmids. The nucleus enrichment of GFP-LC3 is independent of the autophagy induction (Kimberly R. Drake, et, al. 2010 Plos One). Besides, only the GFP-LC3 dots per cell were considered as the marker of autophagosomes accumulation (Daniel J Klionsky, AUTOPHAGY, 2016, VOL. 12, NO. 1, 1-222).

Moreover, we performed the endogenous LC3B immunofluorescent staining experiment. The results consistently showed that ENDOG overexpression promoted the LC3B dots

accumulation. Additionally, considering the disadvantage of the over-expression system, we also evaluated the function of ENDOG in autophagy by knocking out ENDOG or the ENDOG specific inhibitor treatment by using multiple assays. All the results consistently indicated that ENDOG promoted autophagy.

(B) Immunofluorescent staining of endogenous LC3B in L02 cells. (L02 with no transfection or transfected with pK-Myc or ENDOG for 48 hours; Scale bar = 10 μ m).

Fig. 2A: authors claimed "showed decrease LC3B accumulation". Again images show no difference, but statistically analysis with significance, which is strange and brought concerns. The conclusion is not convincing.

Answer: We tested several mice livers, and the results consistently showed that the level of LC3B-II was reduced significantly in *Endog*^{-/-} mice compared with *Endog*^{+/-} mice under the starvation treatment. To be convenient, we showed the normalized grayscale value of LC3B-II blots as below.

Moreover, the electron microscopic results also showed less autophagic vesicles (autolysosomes + autophagosomes) in *Endog^{-/-}* mice livers under the starvation treatment (Figure 2C-D). All these data suggested that loss of ENDOG repressed starvation-induced autophagy in mice livers.

Fig. 3J: IgG controls are missing.

Answer: We did the IgG control in those experiments, but we didn't show the IgG control blot in the previous submission due to the figure size limitation. As requested, we present the data with IgG in the revised submission. The raw image with this IgG control was provided in source data in our previous submission.

For the previous comment “The link between DNA damage response and endoG-mediated autophagy was rather weak. It was not clear what was the rationale to use etoposide to damage DNA and how it could be linked to starvation-induced autophagy. It was not a surprise to see the involvement of endoG in DNA damage response according to previous studies. However, this study did not show how endoG was involved in DNA damage response. Authors claimed that endoG mediated DNA damage response by its endonuclease activity. However, there is no any direct evidence to support this conclusion. It is mandatory to use its endonuclease deficient mutant to confirm their conclusion.”, authors indeed provided substantial data to confirm the effects of endoG on DNA damage response. However, no direct convincing data were provided in the revision to link between endoG-mediated DNA damage response with endoG-mediated autophagy.

Answer: In our previous submission, we demonstrated that at the early point of starvation, ENDOG promoted DNA damage which may activated PARP-1/AMPK and repressed the mTOR pathway and finally promoted autophagy (Figure 5). Moreover, using the ATM inhibitor KU60019 to block the DNA damage response, we found ENDOG-mediated DNA damage and autophagy were abolished (Figure 6 H-K). These data suggested a strong link between ENDOG-mediated DNA damage and autophagy.

Figure S14 E should be presented on the same blot.

Answer: The results were run on the same blot as showed in the original raw data in previous manuscript. But, we repeated this experiment again and showed the new blots as requested.

REVIEWERS' COMMENTS

Reviewer #1 (Remarks to the Author):

The author answered all my questions.

Point-by-point response

We appreciate Reviewer #1 again for his valuable advice and suggestions for improving this manuscript, and eventually satisfied with our work. Below are our point-by-point responses (*blue italic font*) to the reviewers' comments.

REVIEWERS' COMMENTS

Reviewer #1 (Remarks to the Author):

The author answered all my questions.

Thank you!